# Cervical epithelial damage promotes *Ureaplasma parvum* ascending infection, intrauterine inflammation and preterm birth induction in mice

Ioannis Pavlidis[1]*, Owen B. Spiller [2]*, Gabriella Sammut Demarco[1], Heather MacPherson[1],
Sarah E.M. Howie [3], Jane E. Norman[4] & Sarah J. Stock [1,5]*

Around 40% of preterm births are attributed to ascending intrauterine infection, and *Ureaplasma parvum* (UP) is commonly isolated in these cases. Here we present a mouse model of ascending UP infection that resembles human disease, using vaginal inoculation combined with mild cervical injury induced by a common spermicide (Nonoxynol-9, as a surrogate for any mechanism of cervical epithelial damage). We measure bacterial load in a non-invasive manner using a luciferase-expressing UP strain, and post-mortem by qPCR and bacterial titration. Cervical exposure to Nonoxynol-9, 24 h pre-inoculation, facilitates intrauterine UP infection, upregulates pro-inflammatory cytokines, and increases preterm birth rates from 13 to 28%. Our results highlight the crucial role of the cervical epithelium as a barrier against ascending infection. In addition, we expect the mouse model will facilitate further research on the potential links between UP infection and preterm birth.

[1] Tommy's Centre for Maternal and Fetal Health at the MRC Centre for Reproductive Health, Queen's Medical Research Institute, University of Edinburgh, 47 Little France Cresent, Edinburgh EH16 4TJ, UK. [2] Division of Infection and Immunity, School of Medicine, Cardiff University, 6th floor University Hospital of Wales, Cardiff CF14 4XN, UK. [3] MRC Centre for Inflammation Research, Queen's Medical Research Institute, University of Edinburgh, 47 Little France Cresent, Edinburgh EH16 4TJ, UK. [4] Faculty of Health Sciences, University of Bristol, 5 Tyndall avenue, Bristol BS8 1UD, UK. [5] Usher Institute, University of Edinburgh, NINE Edinburgh BioQuarter, Edinburgh EH16 4UX, UK. *email: Ioannis.Pavlidis@ed.ac.uk; SpillerB@cardiff.ac.uk; Sarah.stock@ed.ac.uk

Preterm birth (PTB) is defined by the World Health Organisation (WHO) as any birth prior to 37 completed weeks of gestation[1] ranging from 5 to 18% of all pregnancies internationally[2]. PTB is the cause of 35% of the annual 3.1 million global neonatal deaths; complications of PTB are the second most common cause of death in children under 5 years old; and PTB carries a significant burden of morbidity and disability for survivors[3]. The potential aetiologies of PTB are diverse[4]; however, at least 40% of PTB is attributed to intrauterine infection[5]. From a mechanistic perspective, infection can trigger PTB through a cascade of events. First, the microbial structures and products are recognised by Toll-like receptors (TLRs) that are expressed in the decidua[6], the myometrium[7], the foetal membranes[8] and the placenta[9] during pregnancy. Following receptor ligation, downstream intracellular events driven by the transcription factors nuclear factor-κB (NF-κB)[10] and activator protein-1 (AP-1)[11] lead to the increased expression of pro-inflammatory cytokines, among which interleukin-1b (IL-1b)[12], IL-6[13], IL-8[14] and tumour necrosis factor-α (TNFα)[15]. These cytokines can stimulate the expression of the prostaglandin-synthesising enzyme cyclooxygenase-2 (COX-2)[16] as well as the expression of matrix metalloproteinases (MMPs)[17]. Prostaglandins are known to initiate co-ordinated myometrial contractions and along with MMPs they facilitate membrane rupture and cervical ripening[18,19].

During pregnancy, the cervix remains closed and rigid, providing structural support to the uterine contents. At the same time, it constitutes a physical and functional barrier that protects the foetus against ascending infection from the bacteria-rich vagina. The cervical epithelium is critical for this role, as it has the ability to sense a microbial challenge by expressing TLRs[20] and stimulate the innate and adaptive immune system by producing pro-inflammatory cytokines and antimicrobial peptides such as SLPI[21], Elafin[22] and LL-37[23]. It also contributes to the physical barrier function during pregnancy through a strict temporal regulation of the expression of tight junction proteins[24] and desmosomes[25]. Excisional procedures for the treatment of cervical intraepithelial neoplasia (CIN) where cervical epithelium is removed, along with underlying stroma, are commonly performed and have been strongly associated with pregnancy complications including PTB, yet the underlying mechanisms remain elusive[26]. However, the integrity of the cervix can also be challenged by less invasive events, including alterations to the vaginal flora, viral infection and exposure to chemicals.

*Ureaplasma* spp. are the most common organisms isolated from amniotic fluid obtained from women who present with the PTB antecedents of preterm labour with intact membranes; preterm premature rupture of membranes (pPROM); short cervix associated with microbial invasion of the amniotic cavity; as well as from infected placentas[27]. Furthermore, viable *Ureaplasma* can be cultured from 23% of cord bloods obtained from preterm neonates[28], and a recent human placenta microbiome study reported that out of all the bacteria that they found in human placentas, *Ureaplasma* spp. along with the less common *Streptococcus anginosus* are significantly associated with PTB[29]. Generally considered a low virulence organism, *Ureaplasma* spp. has long been linked to chorioamnionitis[30] that predisposes to PTB[31] and foetal injury[32]. Despite repeated reports of *Ureaplasma* association with infertility, early pregnancy loss, stillbirth, PTB and neonatal morbidities, it is also true that *Ureaplasma* spp. can be detected in vaginal flora in 40–80% of healthy women[27]; therefore, not all lower urogenital tract infections lead to PTB. Consequently, it is probable that more than one 'insult' is required to potentiate the likelihood of PTB. Although systematic investigation may yet discover virulence genes specific to PTB-associated clinical *Ureaplasma* strains, experimental *Ureaplasma parvum* (UP) infections of pregnant macaques with the same prototype strain have been reported to have conflicting outcomes for PTB, suggesting that other experimental design differences may have potentiated the pathogenicity of *Ureaplasma*[33,34] However, none of these experimental *Ureaplasma* infection models address ascending infection from the vagina through the cervix to the uterus as the most common route of a PTB-triggering infection in humans[35].

In this report, we examine a mouse model of ascending infection following vaginal inoculation by UP, and characterise an increased rate of ascending infection and PTB resulting from pre-infection cervical exposure to N-9, a commonly used spermicide that has been approved for use by the US Food and Drug Administration.

## Results

**A model of cervical epithelial damage in mouse pregnancy.** A model of cervical epithelial insult was developed by exposing pregnant mice (day 17 gestation of 21-day term pregnancy) to the pharmacological spectrum of N-9 concentrations (2%, 5%, 10% (v/v) in phosphate-buffered saline (PBS)) representative of those used in spermicidal human contraceptives. Pregnant C57Bl6/J mice received a 60 μL intravaginal bolus of N-9 diluted in PBS or PBS control and the cervices harvested for investigation 8 h later. Paraffin sections of the cervix were stained with Alcian Blue, followed by periodic acid/Schiff's reagent and a pathological score for epithelial integrity determined for the cervix (scoring system demonstrated in Fig. 1a). Exposure to 2%, 5% and 10% N-9 resulted in statistically significant increases in pathological score (Fig. 1b; $4.33 \pm 0.36$, $P = 0.0043$ for 2%; $3.89 \pm 0.29$, $P = 0.0265$ for 5%; and $4.42 \pm 0.48$, $P = 0.0033$ for 10%, mean ± standard error of the mean (SEM), Dunnett's multiple comparisons test) compared to PBS controls ($2.083 \pm 0.37$) (weighted Cohen's $\kappa = 0.952$). Pathological scores for vaginal epithelia examined from the same animals were similarly increased in the N-9-treated group (Supplementary Fig. 1; $4.08 \pm 0.37$, $P = 0.0075$ for 2%; $3.56 \pm 0.95$, $P = 0.0434$ for 5%; and $4.42 \pm 0.34$, $P = 0.0033$ for 10%, Dunnett's multiple comparisons test) compared to PBS controls ($1.54 \pm 0.34$) (weighted Cohen's $\kappa = 0.774$). Further evidence of insult to the cervix was determined by quantifying neutrophil infiltration into the cervical epithelium (Fig. 2a) and cervical stroma (Fig. 2b) by enumerating Ly-6G-stained neutrophils in haematoxylin-counterstained paraffin sections. Only the highest dose of 10% N-9 resulted in a statistically significant increased neutrophil infiltration score in the cervix of pregnant mice (Fig. 2c; $2.75 \pm 0.75$, $P = 0.1234$ for 2%; $1.33 \pm 0.33$, $P = 0.962$ for 5%; and $3.75 \pm 0.75$, $P = 0.0147$ for 10%, Dunnett's multiple comparisons test) compared to PBS control ($1 \pm 0$) (Fig. 2c) (weighted Cohen's $\kappa = 0.874$). These findings were similar for neutrophil infiltration in the vaginal epithelium (Supplementary Figs. 2 and 3; $4.75 \pm 0.63$, $P = 0.0115$ for 2%; $3.67 \pm 1.45$, $P = 0.1046$ for 5%; and $5.5 \pm 0.5$, $P = 0.0032$ for 10% compared to PBS control $1.25 \pm 0.25$, Dunnett's multiple comparisons test) (weighted Cohen's $\kappa = 0.849$).

N-9 induction of cell proliferation as an indirect indication of cervical damage was assessed by quantifying the percentage of proliferating basal cells (visualised by Ki-67 staining of paraffin sections) in a total area of at least 1 mm per tissue section of the cervical basement membrane (representative images shown in Fig. 3a). Induction of increased cell proliferation, likely as a repair response to tissue injury, was found for all three N-9 concentrations (Fig. 3b; $42.21 \pm 2.96$, $P = 0.0437$ for 2%; $43.94 \pm 2.66$, $P = 0.0298$ for 5%; and $50.46 \pm 3.47$, $P = 0.0011$ for 10%, Dunnett's multiple comparisons test) relative to PBS controls ($31.73 \pm 1.2$). However, no increased cell proliferation was observed for the vaginal epithelium (Supplementary Fig. 4).

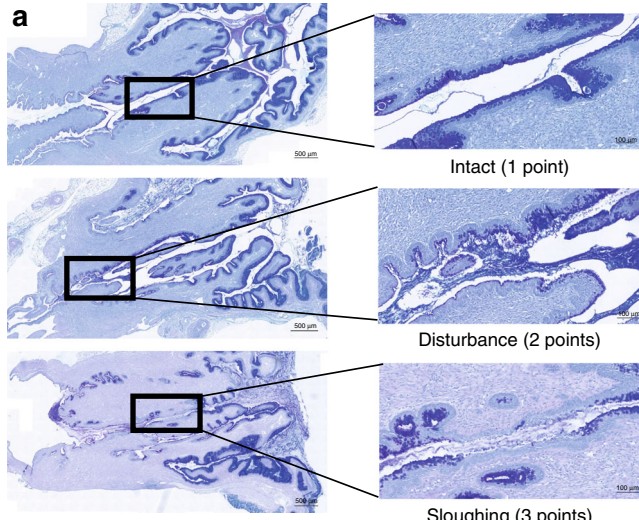

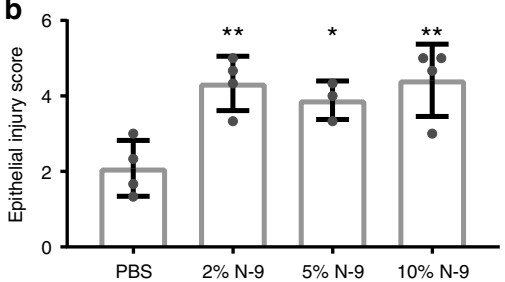

**Fig. 1 Intravaginal N-9 disrupts cervical epithelial morphology during pregnancy.** In the morning of D17 of gestation, mice received either N-9 (2% $n = 4$, 5% $n = 3$ or 10% $n = 4$ in PBS) or PBS control ($n = 4$) via intravaginal inoculation. After 8 h, mice were sacrificed for tissue collections. Cervical tissue sections were stained with AB/PAS and a morphological damage scoring system was used to assess epithelial damage (**a**). N-9 significantly damages the morphology of the cervical epithelium during pregnancy (**b**). Error bars indicate SD. Statistical significance was assessed using one-way ANOVA with Dunnett's multiple comparisons test against PBS group (**$P < 0.005$ for 2% N-9 and 10% N-9 vs. PBS, *$P < 0.05$ for 5% N-9 vs. PBS). Source data are provided as a Source Data file.

**N-9 does not affect timing of delivery or pup survival.** To investigate whether the N-9-induced cervicovaginal toxicity induced PTB or pup mortality, the above experiments were repeated in pregnant mice (precisely determined to be the morning of day 17 gestation based on timed mating/observed cervical plug formation) along with a higher dose of 40% (v/v) to capture the whole spectrum of a dose response. Delivery time was defined as the emergence of the first pup (determined by review of continuous time-stamped CCTV recording) in hours post-N-9 administration and the percentages of live-born pups for each mouse were also determined. PBS control animals delivered an average of $57.93 \pm 2$ h post procedure with $98 \pm 2\%$ of pups surviving (Supplementary Fig. 5). Time to delivery and pup survival were similar in animals exposed to 2–40% N-9 (time to delivery $60.05 \pm 4.97$ h, $P = 0.985$ for 2%; $58.1 \pm 3.58$ h, $P = 0.999$ for 5%; $64.34 \pm 4.18$ h, $P = 0.49$ for 10%; and $68.3 \pm 3.66$ h, $P = 0.2$ for 40% N-9; pup survival $97.8 \pm 2.2\%$, $P > 0.999$ for 2%; $98 \pm 2\%$, $P > 0.999$ for 5%; $96.75 \pm 2.14\%$, $P > 0.999$ for 10%; and $90.6 \pm 6.49\%$, $P = 0.9486$ for 40% N-9, Dunnett's multiple comparisons test).

**Cervical damage facilitates ascending UP infection in pregnancy.** To investigate whether N-9-induced cervical epithelial challenge predisposes to ascending infection during pregnancy, we examined the magnitude and frequency of ascending bacterial infection following experimental vaginal inoculation of UP. UP was chosen as this is the most frequently isolated bacterial species associated with PTB[36] and the UP strain (HPA5) utilised has extensively been used in sheep models of intrauterine infection[37–39].

However, our UP strain used in these studies was modified by insertion of the Promega Nanoluc® luciferase gene to enable non-invasive in vivo imaging/quantification of UP infection (Supplementary Fig. 6), as it has been successfully used previously for imaging *Trypanosoma cruzi* infection progression in mice over 126 days[40]. We would not anticipate NanoLuc gene insertion and expression to alter bacterial virulence and have no evidence that it does. In vivo UP infection was imaged by bioluminescence signal intensity (BLI) imaging of the whole abdomen. For ascending infection studies, pregnant mice were treated with 10% (v/v) N-9 or PBS alone in the afternoon of day 16 gestation, to allow recovery prior to intravaginal inoculation of $2.5 \times 10^6$ infectious units of UP-Luciferase, or sterile *Ureaplasma*-selective medium (USM), in the morning of day 17 gestation. The dose of 10% (v/v) N-9 was chosen as it was found to be the only one inducing both morphological damage and neutrophil infiltrations. Visualisation directly after administration of UP-Luciferase incubated with the NanoLuc® substrate furimazine either vaginally or intrauterine under ultrasound guidance generated no signal at all (Supplementary Fig. 7). Twenty-four hours post infection (hpi), the NanoLuc® substrate furimazine was administered intraperitoneally and luciferase signal was quantified by in vivo bioluminescence optical imaging using software-based signal quantification of the abdomen (pre-defined region of interest).

Bioluminescence signals above background (set by PBS control animal images) were demonstrated in mice infected with UP and a representative image is shown (Fig. 4a). Mice not infected with UP exhibited only background BLI levels ($706 \pm 48$ counts for control treatments PBS + USM, $734 \pm 55$ counts for epithelial damage alone N-9 + USM) (Fig. 4b). Higher bioluminescence signal, representing increased bacterial load, was observed among N-9-treated, UP-infected mice (N-9 + UP; $6.12 \times 10^4 \pm 2.57 \times 10^4$ counts; $P = 0.0131$, unpaired $t$ test on log-transformed bioluminescence values) relative to UP-infected mice pre-treated with PBS (no cervical damage control; PBS + UP) ($9.5 \times 10^3 \pm 7.4 \times 10^3$ counts; Fig. 4b). High-intensity signals formed the expected shape and location of the uterus, showing bulges representing gestating foetuses (Supplementary Fig. 8). Furthermore, the reliable in vitro threshold of detection for UP-Luciferase was found to be 1000 colony-forming unit (CFU)/mL when imaged directly in a luminometer (Supplementary Fig. 9); therefore, high-intensity signal in the in vivo setting that is detected across the mouse abdomen would represent a substantially higher than $10^3$ CFU/mL number of microorganisms.

**Cervical damage results in higher amniotic fluid UP titres.** To assess the capacity of UP to establish an infection at the administration site and whether this was affected by pre-treatment with N-9, we quantified viable UP-Luciferase, by titration in growth medium, flushed from the vagina with PBS at 48 hpi. Inoculated mice were found to have culturable genitourinary levels of UP-Luciferase, ranging from 10 to $1 \times 10^7$ infectious organisms per 15 μL effluent, while none of the uninfected control animals showed growth of UP in vaginal flushes. No differences in vaginal colonisation was observed between UP-infected animals pre-treated with N-9 or PBS ((Fig. 5; $1.3 \times 10^6 \pm 7 \times 10^5$ Colour

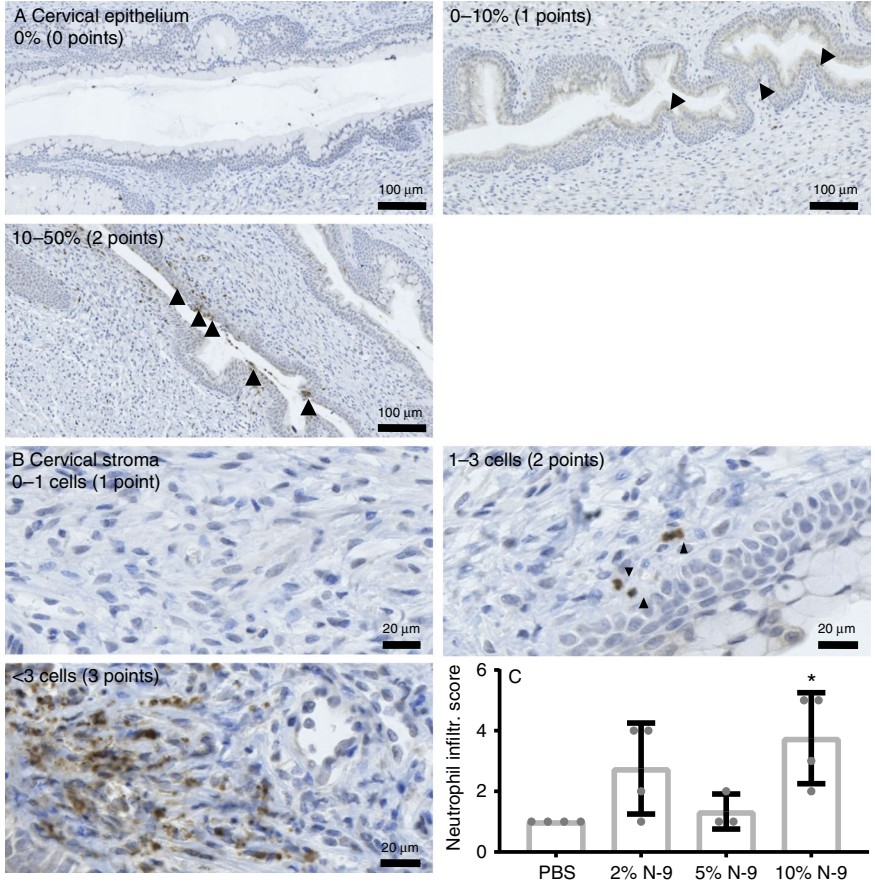

**Fig. 2 Intravaginal N-9 results in neutrophils infiltrations in the cervix during pregnancy.** In the morning of D17 of gestation, mice received either N-9 (2% $n = 4$, 5% $n = 3$ or 10% $n = 4$ in PBS) or PBS control ($n = 4$) via intravaginal inoculation. Eight hours later, mice were sacrificed for tissue collections. Anti-Ly-6G immunohistochemistry on cervical tissue sections was used to assess the presence of neutrophils. A neutrophil infiltration scoring system was used to quantify the presence of neutrophils in the cervical epithelium (**a**) and stroma (**b**), the total score being the sum of the two. Ten per cent N-9 significantly increased the neutrophil infiltrations in the cervix (**c**). Arrowheads indicate positively stained cells. Error bars indicate SD. Statistical significance was assessed using one-way ANOVA with Dunnett's multiple comparisons test against PBS group (*$P < 0.05$ for 10% N-9 vs. PBS). Source data are provided as a Source Data file.

Changing Units (CCU) for PBS + UP; $1.1 \times 10^6 \pm 7 \times 10^5$ CCU of vaginal flush for N-9 + UP)). However, higher UP titres were found in the amniotic fluid surrounding mouse foetuses of N-9-pre-treated infected animals at 48 hpi, with titres (per 15 µL amniotic fluid) of $2.2 \times 10^6 \pm 7 \times 10^5$ CCU for N-9 + UP vs. $7.2 \times 10^4 \pm 3.7 \times 10^4$ CCU for PBS + UP ($P = 0.0356$, unpaired $t$ test with Welch's correction on log-transformed titration values) in proximal gestational sacs and $2.2 \times 10^6 \pm 8 \times 10^5$ CCU for N-9 + UP vs. $3.7 \times 10^3 \pm 2.8 \times 10^3$ for PBS + UP; $P = 0.0466$, unpaired $t$ test with Welch's correction on log-transformed titration values) for distal sacs (Fig. 5).

**Cervical damage results in higher tissue *ureC* messenger RNA (mRNA) copy number.** Molecular methods of quantifying ascending UP infection were used to complement the culture data. Messenger RNA levels for the essential gene *ureC* were used to quantify active bacterial gene transcription in frozen tissue samples isolated from the placenta, myometrium, foetal membranes and foetal lung tissues of mouse foetuses proximal and distal to the cervix harvested at 48 hpi. *ureC* mRNA was not detected in any of the tissues from mice that did not receive intravaginal UP. Pre-treatment with N-9 followed by UP infection resulted in a higher *ureC* copy number in the placenta (Fig. 6; $29,490 \pm 5037$ copies; $P = 0.0093$, unpaired $t$ test on log-transformed values) and in the foetal membranes (Fig. 6c;

$48,270 \pm 7716$ copies; $P = 0.048$, unpaired $t$ test on log-transformed values) in proximal gestational sacs compared to UP-infected mice pre-treated with PBS control only ($10,386 \pm 2862$ copies for placenta; $22,602 \pm 8623$ copies for foetal membranes). No differences in *ureC* gene copy number were observed in the foetal lung of foetuses in proximal gestational sacs ($333.8 \pm 190.1$ copies for N-9 + UP vs. $218.8 \pm 108.2$ copies for PBS + UP), or in tissues from mouse foetuses in distal gestational sacs, for N-9-pre-treated UP-infected animals compared to PBS-pre-treated controls (Fig. 6).

**Increased PTB rates on mice treated with N-9 followed by UP.** The normal time to delivery for C57Bl/6J mice from the morning of day 17 gestation is ~60 h; therefore, we set parameters for induction of PTB as delivery ≤48 h post-bacterial inoculation, which was performed in the morning of day 17 (consistent with previous studies examining LPS-induced PTB[41,42]). PTB rates for N-9-pre-treated UP-infected animals (N-9 + UP) were increased (10/36; 28%; $P = 0.0104$, Fisher's exact test) compared to animals receiving PBS and sterile USM (Fig. 7; PBS + USM = 0%). PTB was observed in 1/15 uninfected animals pre-treated with N-9 in the absence of UP infection (N-9 + USM = 6%; $P = 0.4571$, Fisher's exact test) and in 4/30 UP-infected animals in the absence of N-9 treatment (PBS + UP = 13%; $P = 0.1476$, Fisher's exact test). Sixty-three per cent (62/98) of the spontaneously

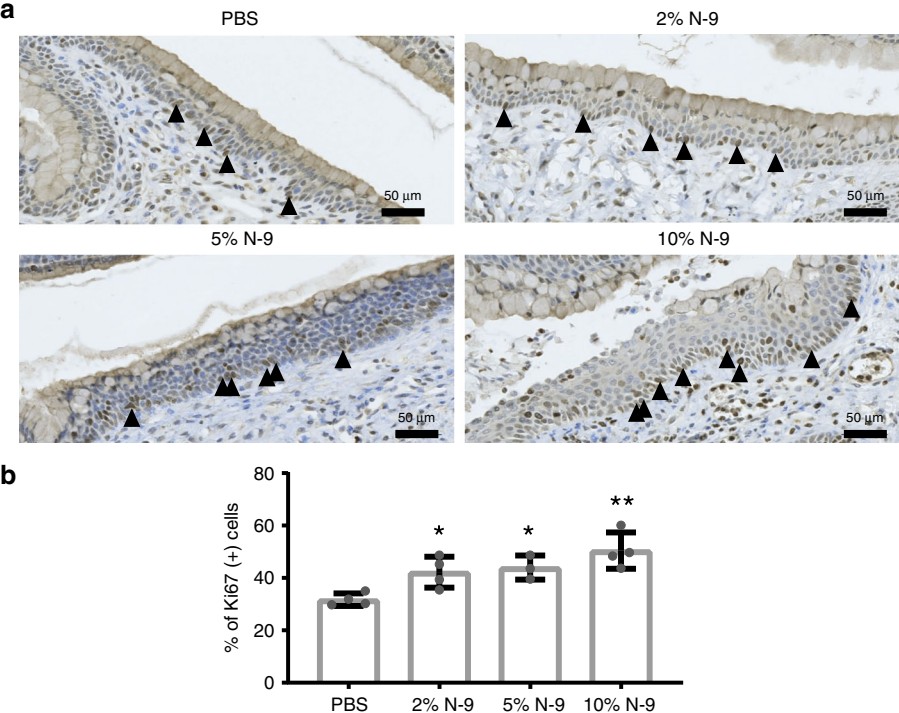

**Fig. 3 Intravaginal N-9 increases basal cervical epithelial cell proliferation during pregnancy.** In the morning of D17 of gestation, mice received either N-9 (2% $n = 4$, 5% $n = 3$ or 10% $n = 4$ in PBS) or PBS control ($n = 4$) via intravaginal inoculation. At 8 h, mice were sacrificed for tissue collections. Anti-Ki-67 immunohistochemistry on cervical tissue sections was used to assess cellular proliferation at the basal layer of the cervical epithelium. Representative images are shown in **a**. The percentage of Ki-67-positive cells was calculated across an area covering at least 1 mm of the basal layer. N-9 significantly increases cellular proliferation at the cervix basal layer (**b**). Error bars indicate SD. Statistical significance was assessed using one-way ANOVA with Dunnett's multiple comparisons test against PBS group (**$P < 0.005$ for 10% N-9 vs. PBS, *$P < 0.05$ for 2% N-9 and 5% N-9 vs. PBS). Source data are provided as a Source Data file.

preterm born pups (vaginally delivered) were found alive during twice daily cage checks. However, all preterm pups were culled on discovery in line with licence requirements and no investigations were performed on preterm animals.

**Gestational tissue cytokine response after ascending UP infection.** There is a well-established link between infection in the uteroplacental tissues and a subsequent inflammatory response driving a premature activation of the labour cascade. To explore the effects of UP ascending infection on inflammation, protein levels (enzyme-linked immunosorbent assay (ELISA)) of TNFα and (C–X–C motif) ligand 2 (CXCL-2), and mRNA expression (quantitative PCR (qPCR)) of the pro-inflammatory cytokines TNFα, IL-1β, CXCL-1, CXCL-2 or IL-6 were examined in the placenta, myometrium, foetal membranes and foetal lung in frozen tissue collected at 48 hpi.

When compared to control (PBS + USM), UP infection without prior cervical damage (PBS + UP) increased the protein expression of cytokine TNFα in myometrium adjacent to proximally located gestational sacs, but did not increase TNFα or CXCL-2 protein expression in any of the other tissues examined (Fig. 8). In contrast, when preceded by cervical damage, UP treatment (N-9 + UP) increased levels of CXCL-2 and TNFα in the placenta of proximally and distally located gestational sacs; CXCL-2 and TNFα in the myometrium adjacent to proximal sacs; and CXCL-2 in the foetal membranes of proximal sacs (Fig. 8).

Fold changes in the mRNA expression of pro-inflammatory cytokines relative to the control group (PBS + USM) are shown in Fig. 9. An increased inflammatory response was observed in gestational tissues in response to UP infection with (N-9 + UP) and without (PBS + UP) cervical damage. This was reflected in

increases in the mRNA levels of TNFα, IL-1β, CXCL-1, and CXCL-2, myometrium and foetal membranes of the gestational sacs most proximal to the cervix (Fig. 9). Changes were less prominent at the distal sacs. Cervical damage followed by UP infection (N-9 + UP) was the only treatment that resulted in increased mRNA expression of at least two cytokines in each of the tissues examined (Fig. 9). mRNA levels of TNFα, IL-1β, CXCL-1 and CXCL-2 in the placenta, myometrium and foetal membranes were strongly correlated to bacterial *ureC* mRNA levels (which correspond to active growth of UP; range of Spearman's coefficients of $R^2 = 0.71$ to $0.87$) irrespective of proximal or distal position (Table 1). No intergroup differences in pro-inflammatory cytokine expression were seen in the foetal lung (results not shown).

## Discussion

We have shown that cervical epithelial damage facilitates ascending UP infection into the uteri of pregnant mice, with accompanying PTB and elevation of pro-inflammatory cytokines in the myometrium, foetal membranes and placenta. Our findings further support the role of the vaginal commensal UP in causing pregnancy complications by demonstrating its potential to ascend to the gravid uterus and induce intrauterine inflammation. Importantly, when preceded by cervical damage, UP infection elicits a more robust intrauterine inflammatory response, which significantly increases the risk for PTB. This synergistic effect between the two insults is in line with current consensus towards PTB being a multifactorial syndrome[43]. Our results also highlight the importance of the cervical epithelium in providing a protective barrier against infections in pregnancy, and provide additional rationale for the mechanisms underlying previous

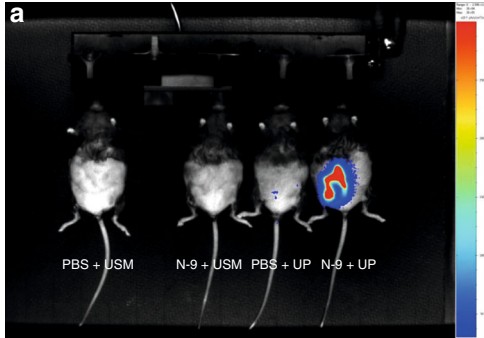

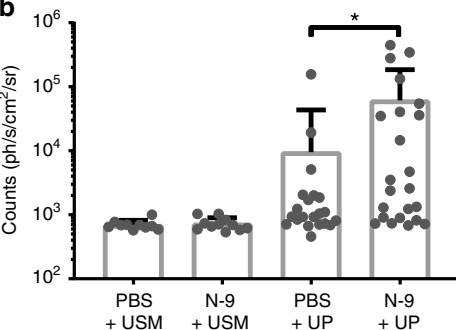

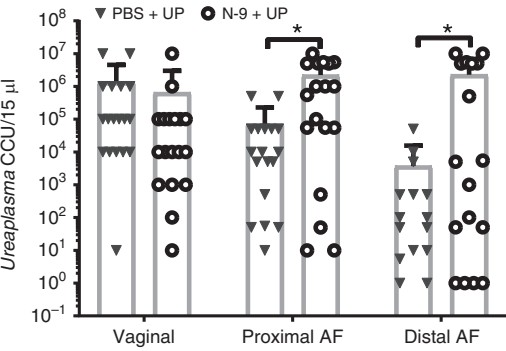

**Fig. 5 Higher UP levels in the amniotic fluid in the presence of cervical damage.** In the afternoon of D16 of gestation, mice received either 10% N-9 in PBS or PBS control via intravaginal inoculation. After 16 h, in the morning of D17 of gestation, mice received either *Ureaplasma parvum* (UP) in ureaplasma-selective medium (USM) or USM control via intravaginal inoculation. After 48 h, in the morning of D19, vaginal flushes and amniotic fluid (AF) from proximal and distal sacs were cultured in USM and UP titres were calculated using the microplate method. There was a significant increase in UP titres at the proximal and distal sites in the amniotic fluid of mice pre-treated with N-9 ($n = 18$) compared to pre-treatment with PBS ($n = 18$). No bacterial growth was detected in uninfected mice. Each dot in the dot plots represents the average of left and right horn for each mouse at either site. Error bars indicate SD. Statistical significance was assessed using unpaired $t$ test with Welch's correction on the log-transformed values of *Ureaplasma parvum* titres for PBS + UP vs. N-9 + UP at all three sites (*$P$ < 0.05; $P = 0.0356$ for amniotic fluid proximal, $P = 0.0466$ for amniotic fluid distal). Source data are provided as a Source Data File.

**Fig. 4 Increased BLI signal by vaginal UP in the presence of cervical damage.** In the afternoon of D16 of gestation, mice received either 10% N-9 in PBS or PBS control via intravaginal inoculation. After 16 h, in the morning of D17 of gestation, mice received either luciferease-expressing UP in ureaplasma-selective medium (USM) or USM control via intravaginal inoculation. After 24 h, in the morning of D18, the luciferase substrate Furimazine was injected intraperitoneally to allow UP localisation. A representative image is shown **a**. Bioluminescence signal was quantified in a region of interest (ROI) covering each mouse's abdomen. Bioluminescence signal coming from luciferase-expressing UP was significantly increased in mice that have been pre-treated with N-9 ($n = 23$) compared to PBS controls ($n = 22$) (**b**). Error bars indicate SD. Statistical significance was assessed using unpaired $t$ test on log-transformed values of bioluminescence quantification for PBS + UP vs. N-9 + UP (*$P = 0.0131$) ($n = 10$ for PBS + USM, $n = 11$ for N-9 + USM). Source data are provided as a Source Data file.

associations between excisional treatment for carcinoma in situ (CIN) and PTB. Cone biopsy or large loop excision of the transformation zone involve removal of part of the cervical epithelium and underlying stroma and are associated with high risk of PTB[26,44]. Increased risk of PTB with larger or repeated excisions has previously been reported[45], but most studies have focused on the mechanical importance of the cervix for providing structural support, ignoring the potential role of the epithelium[46]. Our studies provide empirical evidence of the importance of epithelial integrity.

To generate a cervical epithelial damage model during pregnancy, we used the commonly used spermicide N-9. Our findings of epithelial damage and neutrophil infiltration are in agreement with what has been reported in non-pregnant mice[47,48], rats[49], rabbits[50], pigtailed macaques[51] and humans[52] after vaginal N-9 application. Damage has also been described in different epithelial tissues, such as the monkey rectal tissue[53]. We also found that N-9 doses increased the percentage of proliferating basal cells across the basement membrane, likely a tissue repair mechanism induced by the epithelial injury; Catalone et al.[47] reported a complete regeneration of the epithelium 24 h post treatment in the non-pregnant mouse. Although this seems unlikely given the level of damage reported at 8 h that we saw in the pregnant mouse, it does suggest

that a regeneration process is in place to compensate for the injury being done. In addition, increased proliferation in the cervical epithelium as a response to viral infection with herpes simplex virus type 2 has also been shown to be part of a disorganisation process that compromises the cervical barrier to facilitate ascending infection with *Escherichia coli* in pregnant mice[54].

Our data confirm that N-9 damages the cervical epithelium during pregnancy, but when investigated alone it did not induce PTB or impact foetal mortality. This suggests that cervical epithelial damage in isolation is insufficient to cause PTB. However, ascending genital tract infection and subsequent inflammation are the most common cause of PTB[55]. We found a significant increase in both in vivo bioluminescent imaging of ascending UP infection and increased titres of UP (both in magnitude and frequency of infection reaching mouse foetuses more distal from the cervix) following N-9 cervical epithelial damage. This demonstrates a direct relationship between cervical integrity and ascending infection. We also found a direct correlation in both PTB in mice with ascending infection and significant relationship between UP presence and increased TNFα, IL-1β, CXCL-1 and CXCL-2 cytokine expression in foetal membranes, placenta and the myometrium. This finding is of direct translational importance to the human pregnancy, as *Ureaplasma* spp. are the bacteria most commonly implicated in human PTB[36].

Our results are consistent with two papers both showing that cervical epithelial damage, be it either defective epithelial differentiation[56] or a viral infection[57], increase the risk of PTB following vaginal *E. coli* administration. It is unknown whether the increase in PTB is due to higher rates of infection or higher bacterial titres in the presence of cervical damage; or whether cervical damage itself actively contributes to this phenotype. In our data presented here, it is biologically plausible that the influx of neutrophils into the cervical stroma caused by N-9 at least partly contributed to a pathological premature cervical remodelling as previously described[58].

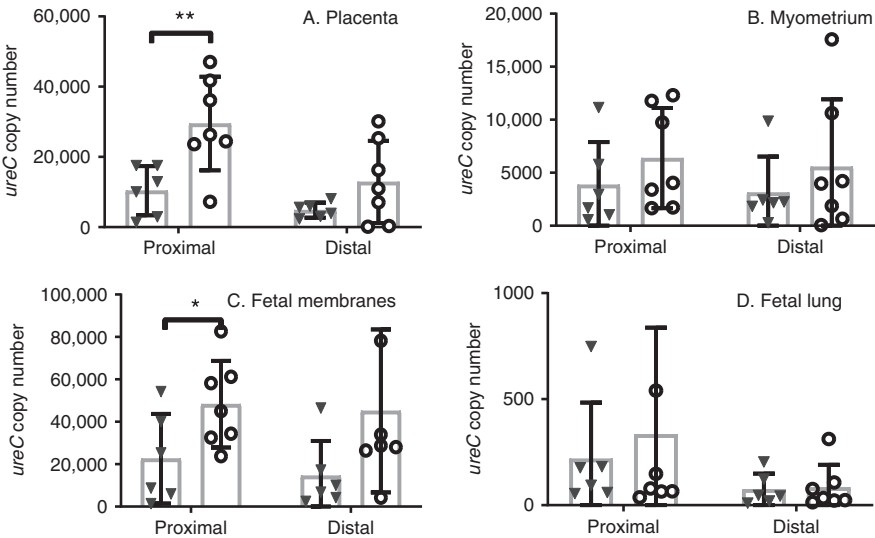

**Fig. 6 Higher UP levels in gestational in the presence of cervical damage.** In the afternoon of D16 of gestation, mice received either 10% N-9 in PBS or PBS control via intravaginal inoculation. After 16 h, in the morning of D17 of gestation, mice received either *Ureaplasma parvum* (UP) in ureaplasma-selective medium (USM) or USM control via intravaginal inoculation. After 48 h, in the morning of D19, gestational tissues were analysed for *ureC* mRNA expression. In mice pre-treated with N-9 ($n = 7$), there was a significant increase in *ureC* copy number at the proximal sites of the placenta and the foetal membranes compared to pre-treatment with PBS ($n = 6$) and a trend increase in all other tissues and sites (**a–d**). No *ureC* copies were detected in uninfected mice. Each dot in the dot plots represents the average of left and right horn for each mouse at either site. Error bars indicate SD. Statistical significance was assessed using unpaired *t* test for PBS + UP vs. N-9 + UP in all four tissues and at both sites (*$P = 0.048$ for N-9 + UP vs. PBS + UP at the foetal membranes' proximal site, **$P = 0.0093$ for N-9 + UP vs. PBS + UP at the placental proximal site). Source data are provided as a Source Data file.

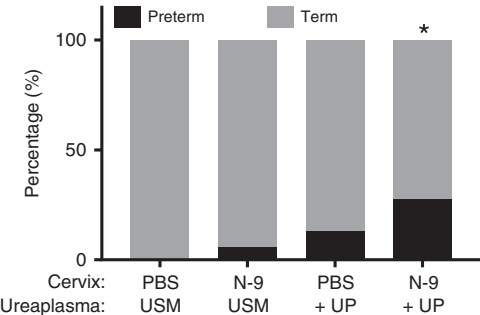

**Fig. 7 Increased PTB rates in the presence of cervical damage.** In the afternoon of D16 of gestation, mice received either 10% N-9 in PBS or PBS control via intravaginal inoculation. After 16 h, in the morning of D17 of gestation, mice received either *Ureaplasma parvum* (UP) in ureaplasma-selective medium (USM) or USM control via intravaginal inoculation. Significantly increased PTB rates compared to controls (defined as delivery of the first pup before up to 48 h after UP administration) was only seen with the combination treatment of N-9 + UP. No bacterial growth or *ureC* copies were detected in uninfected mice. Statistical significance was assessed using Fisher's exact test (*$P = 0.0104$ for PBS + USM vs. N9 + UP). Source data are provided as a Source Data file.

Even if not sufficient to induce PTB, it could accelerate the process driven by UP-induced intrauterine inflammation in PTB models. However, our group has previously published that, in contrast to intrauterine LPS, intravaginal administration of LPS cannot induce PTB[41]. This suggests that cervical inflammation alone is not sufficient for PTB.

Regarding the pathway of infection, Racicot et al.[57] have reported using qPCR that *Ureaplasma urealyticum* can ascend to the uterus and colonise the decidua, and our findings show that closely related species *U. parvum* can pass through the foetal membranes to invade the amniotic fluid.

The role of cervical epithelial integrity is also supported by previous reports showing that inappropriate differentiation of cervical epithelial cells predisposed to ascending infection with *E. coli* at day 16 gestation, resulting in PTB rates of 4/8 for $10^7$ CFU and 3/11 for $10^5$ CFU[56]. Herpes simplex virus-2 infection of the cervical epithelial cells has also been shown to increase PTB rates from 2/6 after a subsequent vaginal administration of $10^5$ CFU *E. coli* to 7/9[54]. Given the small numbers of wild-type animals in both studies, the rates of PTB are comparable to our rates of 4/30 in UP only and 10/36 for N-9 + UP. Our UP infection rates are more representative of *Ureaplasma* spp. associated PTB in patient populations (*E. coli* is not commonly reported to be associated with PTB), and suggests that ascending *Ureaplasma* spp. infection may prove to be a better model for the future. In contrast to *E. coli*, *Ureaplasma* spp. is a very low virulence common vaginal commensal with a prevalence of around 40%[59] and is the most common organism found in PTB. Our PTB rates of 28% are also similar to PTB rates reported in a retrospective study of women with a short cervix and intrauterine *Ureaplasma* spp. infection[60].

Induction of PTB is largely a consequence of an infection-mediated inflammatory response. Therefore, we examined expression of key cytokines in the foetal membranes, placenta and myometrium for infected and uninfected pregnant mice. While no increase in acute inflammatory cytokine IL-6 was found, gene expression levels of TNFα and IL-1β were increased, as were neutrophil-recruiting cytokines CXCL-1 and CXCL-2. Correlating the levels of cytokine expression relative to the levels of UP-derived gene *ureC* expression (a key subunit of the urease enzyme complex) identified a strong and significant relationship in the respective tissues. Increased gene expression was substantiated by increased ELISA measurements for CXCL-2 and TNFα as well. Previous studies, where UP was administered in the intrauterine compartment, have reported similar upregulation of cytokines[61,62]; however, our studies are more relevant to patients, as we have administered UP by ascending infection from the vagina rather than injecting UP or UP-derived lipoproteins

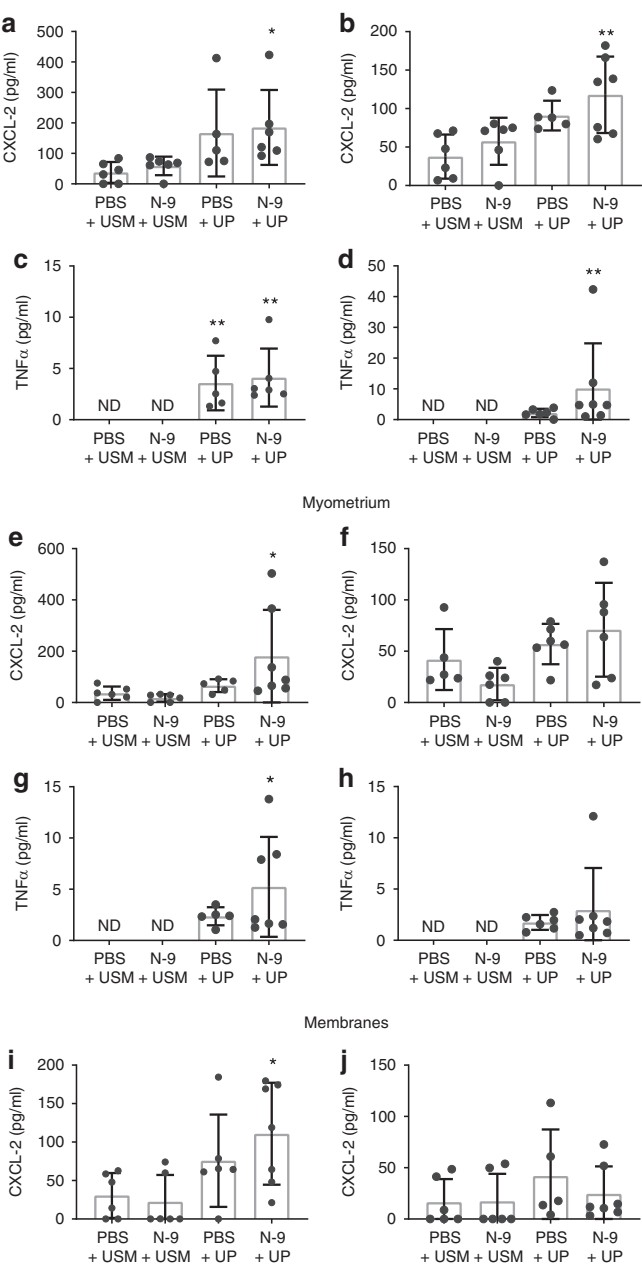

**Fig. 8 Cytokine protein levels in gestational tissues following vaginal UP infection.** In the afternoon of D16 of gestation, mice received either 10% N-9 in PBS or PBS control via intravaginal inoculation. After 16 h, in the morning of D17 of gestation, mice received either *Ureaplasma parvum* (UP) in ureaplasma-selective medium (USM) or USM control via intravaginal inoculation. Tissues were collected 48 h later, protein extracted and analysed using ELISA. Mice that were infected with *U. parvum* than had a previously damaged cervical epithelium (n = 7) demonstrated an increase in the protein levels of CXCL-2 (**a**, **e**, **i**) and TNFα (**c**, **g**) at the proximal site of the placenta, myometrium and membranes compared to vehicle-treated controls (n = 6). At the distal site, there was a difference in the expression of CXCL-2 (**b**) and TNFα (**d**) in the placenta, but not the myometrium or the membranes (**f**, **h**, **j**). Error bars indicate SD. Statistical significance of the treatment effect was assessed using Dunnett's multiple comparisons test against the PBS + USM group for CXCL-2 and Dunn's multiple comparisons test against the PBS + USM group for TNFα (*$P < 0.05$, **$P < 0.01$) (n = 6 for N-9 + USM and PBS + UP) (ND: not detected). Source data are provided as a Source Data file.

In summary, our data highlight the critical role of the cervical epithelium in the barrier function of the cervix against ascending infection and PTB. We also report increased PTB rates after vaginal infection with UP in the mouse. The finding that cervical epithelial damage facilitates ascending vaginal tract infection and amniotic fluid colonisation with UP recapitulates established clinical scenarios and suggests a mechanism explaining why women undergoing excisional treatment of the cervix for CIN are at increased risk for a subsequent preterm delivery. Cervical epithelial integrity appears to be vital for protecting against infection and contributes to a successful pregnancy outcome. While current treatment options for PTB are being proven ineffective, our findings are of significant translational importance in the design of future PTB prevention strategies.

## Methods

**Animal studies**. All animal studies were performed under UK Home Office License 70/8927 (PPL) to J.E.N. and in accordance with the ethical regulations for animal testing and research as set out by the UK Animals (Scientific Procedures) Act of 1986. All proposed experiments and procedures were reviewed against the Home Office Licence and approved by the Named Training and Competency Officer and Named Veterinary Surgeon of the Bioresearch and Veterinary Services of University of Edinburgh. No unexpected adverse events were noted ensuring no breach of the Project Licence Protocols. Specific pathogen-free virgin female C57Bl/6J mice (strain code: 632) were purchased from Charles River Laboratories (Margate, UK) at 6–8 weeks of age. They were housed under tightly regulated temperature (19–23 °C), humidity (55%) and light (12 h dark/light cycles) conditions. Mice were acclimatised for 10 days and then time mated. Confirmation of successful pregnancy was equated to vaginal plug presence at day 1 of pregnancy. Five mice were housed in one conventional cage with regular bedding changes and 24-h access to food and water until the day they would undergo an experimental procedure. Following the procedure, mice were housed in individual conventional cages in an isolated room under CCTV monitoring and under the same husbandry conditions. Pathogen-free status of the housing facility is confirmed by regular cleaning/sterilisation of all rooms with quarterly harvest of sentinel animals in each housing area to confirm the absence of specified pathogen transmission.

***Ureaplasma parvum* culture**. *Ureaplasma parvum* strain HPA5 (sub-strain 137a1, which was recovered from the amniotic fluid of a pregnant sheep following experimental UP infection with HPA5 for 5 weeks) was cultured in USM (Mycoplasma Experience Ltd. Reigate, UK)[69]. Viable bacteria were quantified by 10-fold dilution (series of 15 μL in 135 μL) of inoculation stocks, vaginal flushes or amniotic fluid samples in USM in sterile 96-well plates (Elkay, Basingstoke, UK), sealed with clear adherent sealing film (Elkay, Basingstoke, UK) following incubation at 37 °C for 48 h. For post-infection quantification, all values are calculated for bacterial load in the 15 μL of amniotic fluid or vaginal flush, for inoculating stocks values were calculated per mL of stock. A large-scale stock for inoculation was prepared by collecting mid-log phase growth UP after overnight incubation as previously described for experimental in utero pregnant sheep studies[39]. Inoculating stock was divided into 200 μL aliquots and sufficient aliquots were freshly thawed for the required number of mice to be infected in each batch of pregnant

directly into the uterine horn. UP-mediated cytokine upregulation appears to be mediated via TLR signalling. In vitro studies using a range of cells, including human kidney cells[63], cord blood monocytes[64] and human amniotic epithelial cells[65], have shown upregulation of TNFα, IL-1β, IL-6 and IL-8 (the latter being the human ortholog of CXCL-1 and CXCL-2). Use of small interfering RNA to knock down TLR responses to UP in human amniotic epithelial cells have identified a key role for TLR2, TLR6 and TLR9 in cytokine induction. This could be the pathway of *Ureaplasma* spp.-induced PTB, as these cytokines are known stimulators of the labour-inducing prostaglandins $E_2$ and $F_{2a}$ and their synthesising enzyme COX-2[66,16]. Furthermore, induction of several MMPs, known to be important to parturition remodelling of the cervix for birth by cytokines has also been reported[17,67]. Increased amniotic fluid MMP-9 levels have been reported for UP-positive spontaneous PTBs, but not with induced PTB or spontaneous PTB with negative *Ureaplasma* cultures[68].

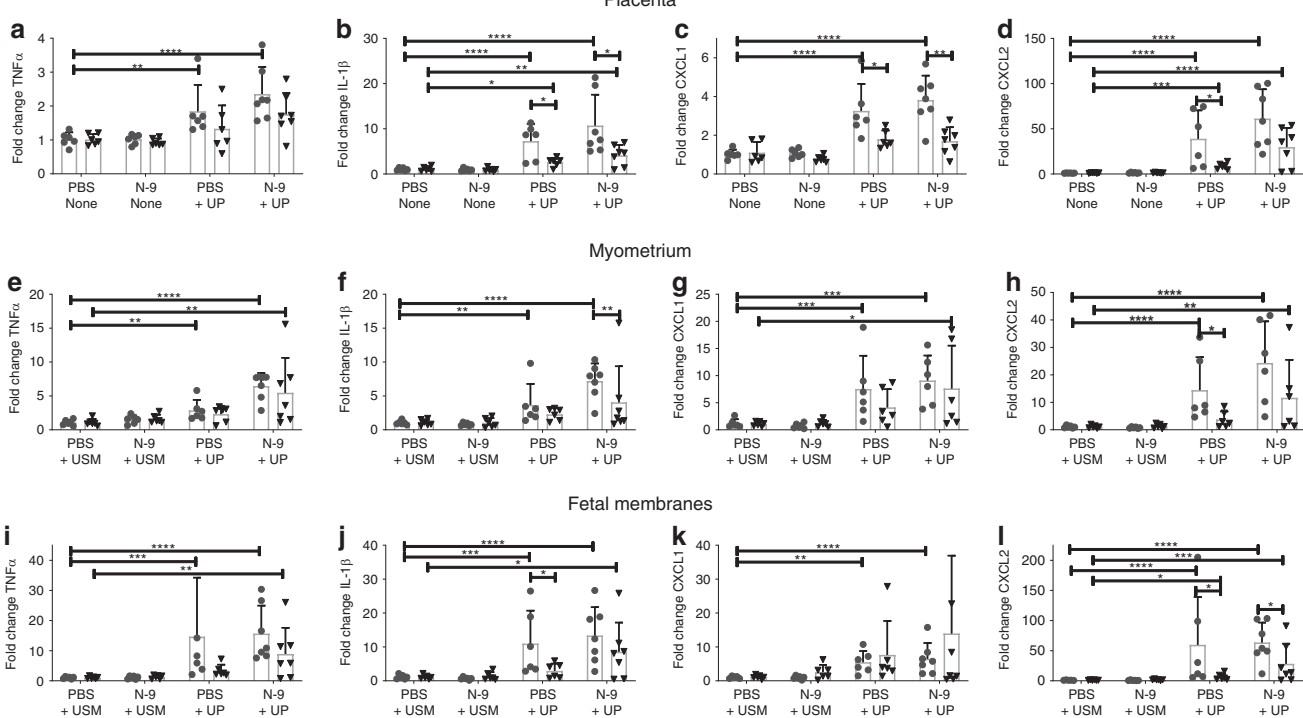

**Fig. 9 Cytokine mRNA levels in gestational tissues following vaginal UP infection.** In the afternoon of D16 of gestation, mice received either 10% N-9 or PBS control via intravaginal inoculation. After 16 h, in the morning of D17 of gestation, mice received either UP or USM control via intravaginal inoculation. Tissues were collected 48 h later and analysed using RT-qPCR. Mice that were infected with UP (PBS + UP, n = 6; N-9 + UP, n = 7) had an increase in the mRNA levels of TNFα (**a, e, i**), IL-1β (**b, f, j**), CXCL-1 (**c, g, k**) and CXCL-2 (**d, h, l**) compared to vehicle-treated controls (PBS + USM, n = 6). The effect was stronger at the proximal site. Error bars indicate SD. Statistical significance of the treatment effect was assessed using Dunnett's multiple comparisons test on the DDCt values against the PBS + USM group for the proximal and distal site. Statistical significance for the site effect was assessed unpaired $t$ test for proximal vs. distal for each treatment group (*$P < 0.05$, **$P < 0.01$, ***$P < 0.001$, ****$P < 0.0001$) (n = 6 for N-9 + USM). Source data are provided as a Source Data file.

**Table 1 Correlation between *ureC* andgene expression changes.**

| Cytokines | Placenta | Uterus | Foetal Membranes |
|---|---|---|---|
| TNFα | 0.51 ($P < 0.01$) | 0.76 ($P < 0.001$) | 0.84 ($P < 0.001$) |
| IL-1β | 0.72 ($P < 0.001$) | 0.79 ($P < 0.001$) | 0.78 ($P < 0.001$) |
| CXCL-1 (IL-8 homologue) | 0.71 ($P < 0.001$) | 0.80 ($P < 0.001$) | 0.70 ($P < 0.001$) |
| CXCL-2 (IL-8 homologue) | 0.78 ($P < 0.001$) | 0.84 ($P < 0.001$) | 0.87 ($P < 0.001$) |
| IL-6 | 0.2 | 0.02 | 0.39 ($P < 0.01$) |

Values are (Spearman's $R^2$) with significance values of correlation within parentheses

mice (aliquots were never refrozen). In contrast to the previous sheep studies, the HPA5 strain of serovar 3 *U. parvum* was genetically manipulated to express the Promega NanoLuc® luciferase gene by site-directed mutagenesis with mini-transposon pMT85 using methods previously described[70]. The altered pMT85 plasmid contained two copies of NanoLuc® luciferase, preceded by the promoter from *U. parvum tufA* gene (synthesised by GenScript, Piscataway, NJ) subcloned into the *Xba*I site in pMT85; the sequence of the entire plasmid is available at GenBank (accession number MN275117). Threshold of detection following incubation of UP-Luciferase with Promega Nano-Glo® Live Cell Assay substrate was found to be equivalent to 1000 bacteria in vitro (Supplementary Fig. 9). Image acquisition was tested following incubation of UP-Luciferase with Promega Nano-Glo® Live Cell Assay substrate and associated controls for our in vivo experiments (Supplementary Fig. 10).

**Mouse model of cervical damage.** In the morning of day 17 gestation, mice were anaesthetised by isoflurane inhalation (5% for induction of anaesthesia, 2.5% for maintenance) and a pipette was used for intravaginal administration of either N-9 (60 μl; 2%, 5%, 10% (v/v) in PBS or PBS alone). Mice were randomly assigned to each group. Care was taken to minimise vaginal leakage of administered bolus, and leakage was monitored by examining dry tissue placed under mice during anaesthesia recovery. Only minor leakages (dampening of tissue) were observed. Mice

were returned to individual cages and tissues were harvested for examination at 8 h post treatment.

**Mouse model of cervical damage and ascending infection.** In the afternoon of day 16 gestation, mice were anaesthetised and received either 10% (v/v) N-9 or PBS as above, monitored for leakage during recovery and returned to individual cages. In the morning of day 17 gestation, mice were re-anaesthetised and received a further vaginal inoculation of either UP-Luciferase (40 μl aliquot of $10^8$ CCU/ml in USM) or 40 μl USM alone. Mice were randomly assigned to each group. To reduce the likelihood of cross-contamination, the group of mice that was treated with UP-Luciferase was handled last. Following successful recovery from anaesthesia, mice were put in individual cages to monitor the time to delivery by continual CCTV (including recording for accurate determination of emergence of first pup). Pre-term delivery was determined as birth occurring <48 h post infection (or mock infection) as normal term delivery is known to be ~60 h from this point. Mice that have not delivered by the 48-h mark after administration of UP-Luciferase (completed day 18 of gestation) were deemed as term and were sacrificed for collection of antepartum tissues. Rapid caesarean delivery of near-term foetuses was performed under terminal anaesthesia as specified by our animal licence; therefore, no assessment for foetal viability could be made. During this period, they also received twice daily physical health checks for signs of potential adverse effects

and to check for preterm born mice, which were recorded and immediately culled in line with the requirements of the animal licence.

**In vivo imaging for luciferase**. Animals in the above group were imaged in vivo for ascending infection and determination of bacterial load 24 h after infection with UP-Luciferase. The abdomens of anaesthetised mice were shaved prior to imaging to minimise signal interference through absorbance, scattering or diffraction of light by the mouse fur and 100 μl of the luciferase substrate Furimazine (Nano-Glo® Live Cell Assay System, Promega, Madison, WI, USA) was administered intraperitoneally. Optical imaging was carried out using the facilities of the Edinburgh Preclinical Imaging, College of Medicine and Veterinary Medicine, University of Edinburgh. Specifically, mice were placed in a PhotonIMAGER$^{TM}$ OPTIMA for optical imaging (Biospace Lab, Nesles-la-Vallee, France). Luminescence was quantified by integral photon counting technology based on intensified charge-coupled devices, allowing real-time display of the bioluminescence signal and recording of kinetics information.

**In vivo imaging analysis**. Imaging analysis was performed using the M3 Vision software (Biospace Lab, Nesles-la-Vallee, France). A standardised elliptical region of interest template was applied to the image of each mouse's abdomen, including the lower abdomen and the cervix, and Photo Acquisition software measurements determined the plateau phase of signal kinetics prior to performing quantification analysis. Measurements were provided as photons per second per centimetre square per steradian (ph/s/cm$^2$/sr). This unit normalises for differences in the mouse size and position on the stage as well as the time frame used for the quantification analysis.

**Tissue collections**. For the cervical damage model, the vagina, the cervix and part of the myometrium were collected as a single continuous tissue and, after the surrounding fat was trimmed off, they were immediately transferred to 4% paraformaldehyde for fixation. Tissues were then embedded in paraffin blocks for immunohistochemical analysis. To accurately assess tissue damage, we deployed the following strategy for processing and analysing our samples: three consecutive 5-μm-thick longitudinal sections (used for histology, Ly-6G and Ki-67 immunohistochemistry, respectively) from three different levels of the lower reproductive tract were processed, each level 50 μm deeper than the preceding one. Subsequently obtained scores for each mouse represent the average of the scores for the three different levels.

For the cervical damage and ascending infection model, the following tissues were collected and snap frozen in dry ice from all term dams: vaginal flush (60 μl of PBS were flushed into the vagina right after the mouse was culled) and cervix. Four foetuses were chosen for tissue collections from each term dam, two from each horn: the one closest to the cervix (proximal) and the one furthest from the cervix (distal). The following tissues were collected and snap frozen in dry ice from all term foetuses: amniotic fluid, placenta, myometrium (whole part of uterus surrounding the foetus), foetal membranes and foetal lung. Unless used immediately, tissues were cut in half and stored in −80 °C for qPCR and protein analysis.

**Paraffin section processing**. Slides were deparaffinised in xylene (Cell Path Ltd, Newtown, UK) and rehydrated in sequential immersion in 100%, 95%, 80%, 70% and ethanol. For histological grading, slides were then immersed in 1% Alcian Blue for 10 min. Following this, slides were rinsed in deionised H$_2$O and were then oxidised in an aqueous solution of 0.5% periodic acid for 10–20 min. After a rinse in deionised H$_2$O, slides were treated with Schiff's reagent for the same length of time and then immersed in Harris' haematoxylin (Cell Path Ltd., Newtown, UK), for 30 s. Slides were then placed in Scott's Tap Water Substitute until sections turned blue. For immunohistology, slides were immersed in sodium citrate buffer (10 mM sodium citrate, pH 6.0 in distilled H$_2$O) and heated to sub-boiling using an InstantPot IP-LUX60 6L/6.33Qt Pressure Cooker (InstantPot, Ottawa, Canada) to unmask antigens. Endogenous peroxidases were removed with 3% hydrogen peroxide treatment for 30 min and non-specific binding removed by incubation with 5% (v/v) normal goat serum. Polymorphonuclear neutrophils were stained with purified rat anti-mouse Ly-6G (lymphocyte antigen 6 complex, locus G) (1:500 in blocking solution, BioLegend, San Diego, USA), while proliferating cells were stained with rabbit anti-Ki-67 (ab15580, 1:1000 in blocking solution, Abcam, Cambridge, UK), by overnight incubation at 4 °C. Primary antibody binding was visualised by ImmPRESS horse radish peroxidase (HRP) anti-rat IgG or anti-rabbit IgG HRP-conjugated secondary antibodies (Vector Laboratories, Peterborough, UK), followed by reaction with 3,3′- diaminobenzidine and counterstained with Harris' haematoxylin. Slides were then dehydrated in ethanol followed by xylene and sealed with cover slips.

**Histological imaging analysis**. Imaging of the samples was performed on a Zeiss Axio Scan.Z1 Slide Scanner (Carl Zeiss AG, Oberkochen, Germany). Imaging analysis was performed using the Zeiss ZEN Blue software (Carl Zeiss AG). This allows for individual operator-led analysis of the images obtained by the slide scanner. Specifically, two independent assessors that were blinded to treatment allocation evaluated all images based on the scoring systems described in the

respective Results sections. Individual scores for each mouse represent the average of the scores assigned by the assessors. Inter-rater reliability was assessed using weighted Cohen's κ.

**Quantitative PCR**. Total RNA was extracted from the placenta, myometrium, foetal membranes and foetal lung tissue collected at the 48 h time point by using the RNeasy mini kit (Qiagen, Crawley, UK) following the manufacturer's instructions. Total RNA (500 ng) was reverse transcribed by using the High Capacity cDNA Reverse Transcription kit (Applied Biosystems, Life Technologies Ltd., Paisley, UK). Pre-designed gene expression assays from Applied Biosystems were used to examine the expression of CXCL-1 (Mm04207460_m1), CXCL-2 (Mm00436450_m1), IL-1β (Mm00434228_m1), IL-6 (Mm00446190_m1) and TNFα (Mm99999068_m1). Target gene expression was normalised for RNA loading by using β-actin, and the expression in each sample was calculated relative to the average of the calibrator group (mice treated with PBS + USM). All qPCR analyses were performed on an Applied Biosystems 7900HT instrument. Data were analysed using the 2$^{−ΔΔCt}$ method. ureC gene expression by HPA5-Luciferase in these samples was quantified by using primers (UUP_FP, AAGGTCAAGGTATGGAAGATCCAA and UUP_RP, TTCCTGTTGCCCC TCAGTCT) and probe (UP_HP, (HEX)-TCCACAAGCTCCAGCAGCAATT TG-(BHQ1)), and absolute copy number of mRNA species determined against a standard curve of 10–1,000,000 copies of a plasmid containing the ureC gene (GenBank accession number NC_010503.1) synthesised by GenScript (Piscataway, NJ)

**Enzyme-linked immunosorbent assay**. Total tissue protein was extracted from the placenta, myometrium and foetal membranes tissues collected at the 48 h time point and quantified using a commercial assay and according to the manufacturer's instructions (Bio-Rad, Hercules, USA). Total protein concentration was adjusted to 3 mg/mL for all samples using ELISA reagent diluent. The levels of CXCL-2 and TNFα were measured using ELISA kits and according to the manufacturer's instructions (R&D Systems, Abington, UK).

**Statistical analysis**. Data are expressed as the mean value of individual groups ± standard deviation (SD), as indicated by error bars, and presented as mean ± standard error of the mean (SEM) in the text. Bioluminescence and amniotic fluid titres were log transformed before analysis and the proportion of live-born pups was arc-sin transformed before analysis. Statistical analysis was performed using the GraphPad Prism 7.0 software (GraphPad, San Diego, CA). Statistical significance in differences between experimental groups was assessed by one-way analysis of variance with Dunnett's multiple comparison test or by unpaired t test with Welch's correction on log-transformed values or by Fisher's exact test for proportions. To determine whether the samples in our populations were normally distributed, a D′ Agostino–Pearson omnibus normality test was performed. All statistical tests were performed as two-way analyses. All measurements included in this study were taken from distinct samples.

**Reporting summary**. Further information on research design is available in the Nature Research Reporting Summary linked to this article.

## Data availability

All raw data from the figures of this manuscript are provided as a Data Source File and may additionally be accessed at https://doi.org/10.7488/ds/2719.

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

## Acknowledgements

This work was funded by Tommy's Charity and the Medical Researcch Council. The work was carried out at the MRC Centre for Reproductive Health (MRC Centre Grant MR/N022556/1). S.J.S. is funded by a Wellcome Trust Clinical Career Development Fellowship (209560). We would like to thank all staff of the Biomedical Research and Veterinary Services of the University of Edinburgh. We would also like to thank Prof. Alistair Williams, Prof. Mark Arends and Prof. Simon Herrington from the Division of Pathology, University of Edinburgh for their advice on Histology and Mr. Adrian Thomson from the Edinburgh Preclinical Imaging Facility for his help with imaging acquisition and analysis.

## Author contributions

I.P. and S.J.S. conceptualised the study. I.P., O.B.S., S.E.M.H. and S.J.S. developed the experimental design. O.B.S. generated, sequenced and characterised the microbial strain and oversaw all microbial quantification. I.P. and H.M. collected mouse samples. I.P. performed mouse monitoring. I.P. and G.S.D. performed the experiments. I.P analysed the data. O.B.S., S.E.M.H., J.E.N. and S.J.S. supervised the work. I.P. and O.B.S. drafted the manuscript. I.P., O.B.S., S.E.M.H., J.E.N. and S.J.S. revised the manuscript.

## Competing interests

The authors declare no competing interests.
