## [Peer Review File · Nature Communications]

Reviewers' comments:

Reviewer #1 (Remarks to the Author):

Pavlidis et al. describes a novel preterm birth model of ascending *Ureaplasma*-infection using bioluminescent *Ureaplasma parvum*. This model uses N-9 to induce epithelial damage and promote ascending vaginal infection of *Ureaplasma*.

The following comments would improve the manuscript:

1. Were the bacteria used for experiments in mid-logarithmic phase and were they prepared prior to each experiment? – this is incredibly important as different phases of growth may affect survival and invasion.
2. Do you have any data on the impact of insertion of the luciferase gene on bacterial virulence?
3. Does the media contain antibiotics? Is there an antibiotic resistant gene in this bacterium? Media was used as a control in your pregnancy experiments, does this have any effect on the cervical epithelium?
4. Results, line 73: can you comment on why treatment with 5% N-9 results in lower pathological scores and neutrophil recruitment than 2% N-9?
5. Figure 1: Can you please comment on why PBS could have an effect on pathological scores?
6. Figure 2: Can you please provide representative neutrophil IHC images for each of the groups.
7. Supplementary Figure 4: there is a non-significant trend to increased time to delivery at higher N-9 concentrations (yet with reduced pup survival), can you explain this?
8. Was the Nano-luc expressing bacteria generated in this study? – if so, please can you give further information in the methods regarding the transposon mutagenesis.
9. Line 175; one of the benefits of bioluminescent imaging is the ability to longitudinally image mice – do you have any data on the time course of bacterial infection?
10. Did you image your bacterial-treated mice at the time of administration of bacteria? This is necessary to prove that it's the cervical damage that leads to increased bacterial ascent.
11. Line 187, do you have any representative images showing the uterine bulges/gestating pups? This is not clear in the provided images. Do you have any imaging from individual organs?
12. Figure 5; there are significant amounts of *Ureaplasma* in amniotic fluid, yet non-significant differences in *Ureaplasma* in fetal lungs, can you please comment on this? Do you have any evidence of bacterial infection in live or dead pups?
13. How does the N-9 damage the cervical barrier? Do you have any evidence of pro-inflammatory change or epithelial barrier integrity/permeability? Neutrophil recruitment was increased in the N-9 group, yet following bacterial administration, more bacteria survive – can you comment on this?

Reviewer #2 (Remarks to the Author):

In this manuscript the authors developed an ascending infection model in mice using a human opportunistic pathogen, *Ureaplasma parvum* combined with cervical/vaginal injury induced with Nonoxynol-9. The authors provide evidence that cervical injury facilitates ascending infection with *U. parvum* and that this is accompanied by preterm labor. A strength of the paper is that the authors used a controversial pathogen that is translationally relevant to human disease. The controversy stems from the fact that infection does not always induce adverse pregnancy outcome. Herein, the authors show that cervical injury is critical for *U. parvum* ascending infection and inflammation linked to preterm birth. While this point is of interest to the field of maternal-fetal medicine, there are some concerns that need to be addressed.

Point 1) Introduction: The key feature in this paper is the interaction of Nonoxynol-9 and *U. parvum* on pregnancy outcome. The authors did not provide a careful explanation of *U. parvum* as an opportunist and the confounding issues with its role in adverse pregnancy outcome. Doing so would actually raise the impact of their paper. I recommend the authors deemphasize *U. parvum* in the first paragraph and instead summarize the current views regarding the mechanics of

preterm birth – including the reemerging view that decidual/myometrial inflammation is the main trigger for preterm labor (this is relevant to some of the data presented). Following the paragraph after the cervix I would add a third paragraph that supports the rationale for using *U. parvum* in this study, including that it is viewed as an opportunistic pathogen and that not all upper reproductive tract infections end up with complications (this includes human studies). Moreover, if the authors do a careful review of *U. parvum* related disease, they will find that tissue injury is important in how this organism enhances disease. This actually strengthens the rationale for using this organism in the first place. As an example, the authors cited Novyi et al, who demonstrated that intraamniotic inoculation of *U. parvum* in pregnant macaques induced preterm labor. While this is true, this study was later contradicted by a paper published by Kallapur's research group that showed that the same strain of *U. parvum* administered in the same way in macaques only induced a nominal amount of inflammation with no induction of preterm labor. What was the difference between these two studies? In Novyi's study, macaques had prior surgery with implantation of electrodes to monitor labor, whereas Kallapur's research group did not. Perhaps the surgery induced enough of a stimulus that enhanced the host immune response to *U. parvum*, and this led to preterm labor. There are other examples in the literature to support this view.

Point 2) Methods: the current view is to provide adequate reporting of live animal studies. Please refer to ARRIVE Guidelines for details, most scientific journals provide a link to this. Key details that need to be added in the methods include animal housing. Were there specific pathogen free animals? What were the housing conditions? Were they maintained under barrier maintenance or conventional housing? Necropsy procedures are not clearly described and are confusing. It sounds as if the pregnancy was terminated before actual labor, if so what was the final gestation day? Technically, these are not pups but fetuses. What portion of the myometrium was isolated, the entire uterus or only the placental bed? The placental bed (myometrial portion in direct association with the placenta) has a unique composition of leukocytes not found in the uterus that is not associated with fetal tissue. This needs to be clarified

Point 2) Methods: how were lesion scores actually obtained? A software program was mentioned but the actual procedure used was not described. How many sections from each animal were examined? Were multiple sections evaluated? How did the software calculate percentages that were later assigned a score? Was the analysis done blinded to treatment?

Point 4) Results: Animal numbers should be clearly stated in each experiment. It is hard to tell in some of the figures what the actual sample sizes are for some of these experiments. Please provide animal numbers in figure legends

Point 5) If infected animals were allowed to go to term, what was the survival rate among pups?

Point 6) The text in Figure 7 is unreadable. Font sizes need to be increased so that the figure is legible.

Point 7) Discussion: line 315, The statement "Our findings confirm the role of the vaginal commensal UP in causing pregnancy complications by demonstrating its potential to ascend to the gravid uterus and induce inflammation and preterm birth" I believe this statement needs to be modified to state the importance of N9 in contributing to this since myometrial inflammation and a significant increase in preterm birth was not observed in the UP only group.

Although N9 by itself did not induce preterm labor or inflammatory response in the upper reproductive tract, the initiation of inflammation cascades by the tissue damage may be a contributing factor (think of DAMPS). Somewhere in the discussion this should be addressed especially since many studies have used bacterial LPS as a substitute for infection and get more dramatic results than *U. parvum* infection alone.

Response to reviewers:

Reviewer #1 (Remarks to the Author):

Pavlidis et al. describes a novel preterm birth model of ascending Ureaplasma-infection using bioluminescent Ureaplasma parvum. This model uses N-9 to induce epithelial damage and promote ascending vaginal infection of Ureaplasma.

The following comments would improve the manuscript:

1. Were the bacteria used for experiments in mid-logarithmic phase and were they prepared prior to each experiment? – this is incredibly important as different phases of growth may affect survival and invasion.

We thank the reviewer for bringing up this insightful question. We designed the experimental infection based on our extensive experience with inducing experimental intrauterine Ureaplasma infection in pregnant sheep (1). To minimise mouse to mouse inoculum variability (including variation in phase of growth), we prepared one large batch of mid-log phase growth Ureaplasma that was frozen at -80°C in 200 microliter aliquots. Ten of these aliquots were thawed and the concentration of Ureaplasma confirmed by both colour-change titration in Ureaplasma selective medium as well as ureC qPCR against a 6-log range standard curve with a plasmid containing the ureC target gene. These two methods were within 10% agreement. For each experiment sufficient aliquots were thawed for the required number of mice to be infected (aliquots were never freeze-thawed or reused). We have expanded the methods section to include the fact that a unique aliquot of a common batch of individual aliquots of mid-log phase growth Ureaplasma were used for all experimental infections. However, it is important to note that Ureaplasma parvum has a generation time of 60 minutes and thawed mid-log phase growth Ureaplasma inoculated into the vagina would quickly recover in the favourable environment and re-enter log phase growth shortly after inoculation.

¹ More information on our intrauterine Ureaplasma infection of pregnant sheep research can be found at doi's: 10.3390/nu11050968; 10.1159/000479021; 10.1371/journal.pone.0180114; 10.1111/aji.12599; 10.1128/AAC.03721-14; 10.1128/AAC.03187-14; 10.1016/j.ajog.2014.02.025.

2. Do you have any data on the impact of insertion of the luciferase gene on bacterial virulence?

We are not sure whether the reviewer is asking if (A) the insertion would reduce the virulence of U. parvum, or whether the author is asking (B) if NanoLuc expression would increase the virulence; we will address both scenarios:

- A) As can be seen in Supplementary Figure 5, the insertion of the luciferase gene disrupts the gene UPA1_G0351 which is a site-specific type III restriction-modification methyltransferase. This disrupted gene is part of a cassette of found in the prototype ATCC serovar 1 strain 27813, but not in other prototype *U. parvum* strains. These genes are non-essential and may add methyl groups to block restriction enzyme cleavage of DNA, and we are unaware of any reports of this particular class of methylases being important to virulence in other bacteria.
- B) The expression of the NanoLuc luciferase has been used in vivo for several reports with influenza, reovirus, alphaviruses and *Plasmodium berghei* as well as to track metastasis and treatment of tumour cells (2), no evidence of toxicity or added virulence related to nanoLuc expression has been noted in these previous publications, and a recent Nature methods publication (Yeh et al., 2017 doi: 10.1038/nmeth.4400) evaluated cytotoxicity of modified NanoLuc high expression systems and the luminescent substrates finding no adverse effects on transfected HEK 291T cells.

We have added “We would not anticipate NanoLuc gene insertion and expression to alter bacterial virulence and have no evidence that it does.” to the manuscript to address this point.

² References: NanoLuc alpha-virus doi: 10.3390/v11070584; NanoLuc reovirus doi: 10.1128/JVI.00401-19; NanoLuc Influenza doi: 10.1007/978-1-4939-8678-1_21, 10.1128/JVI.00593-18, NanoLuc *Plasmodium berghei* doi: 10.1186/s12936-016-1291-9

3. Does the media contain antibiotics? Is there an antibiotic resistant gene in this bacterium?

The commercial *Ureaplasma* medium only contains ampicillin in its formula. *Ureaplasma* is a cell wall-less bacteria and as such is inherently resistant to all beta-lactam antibiotics and colistin (Katz B, Schelonka RL, Waites KB. Clin Microbiol Rev 2005; 18: 757–789). The mini-transposon used for NanoLuc insertion contains an *aacA-aphD* aminoglycoside selection gene that gives resistance to gentamicin and kanamycin, and this is used to select the initial successful minitransposon mutants following transformation (fully detailed in the referenced citation Aboklaish et al., 2014 in the Methods section). However, as the minitransposon insertions are completely stable (no transposase is transferred and absence was confirmed in the sequenced strain) gentamicin is not used in the propagation of the strain for in vivo inoculation stocks, nor is it present in the medium used as controls. Extensive testing on multiple *Ureaplasma* strains using this minitransposon insertion system has not shown any loss or intragenomic movement of the inserted exogenous gene (data not shown). Further clarification on this point has been added to the manuscript.

Media was used as a control in your pregnancy experiments, does this have any effect on the cervical epithelium?

We welcome the reviewer's comment regarding a potential damaging effect of the Ureaplasma medium itself on the cervical epithelium. We do not have any direct data examining the effects of the commercial Ureaplasma selective medium on cervical epithelia cells. Although the formula of the medium is proprietary to the supplier (Mycoplasma Experience Ltd), based on known requirements for Ureaplasma growth, it will likely contain 5-10% v/v yeast extract, 10-20% heat inactivated horse serum and 0.01 M urea (approximately the concentration found in urine). Ampicillin is a common additive to cell culture and addition of animal serum is often a requisite component of cell culture; therefore, these components are unlikely to be toxic to the cervical epithelium. Furthermore, Mosser et al., 2013 (doi: 10.1007/s10616-012-9519-1) have reported that addition of 1 g/L of yeast extract was also found to enhance growth and health of CHO cells in vitro, indicating it is not toxic to cells. However, we acknowledge that yeast proteins may act to activate Pathogen Associated Molecular Patterns (PAMPs) receptors, in addition to those that would be inherently activated by the Ureaplasma lipoproteins. However, as N-9 is detergent-like by nature, we would expect that any contribution of the growth medium itself on epithelial damage to be negligible.

4. Results, line 73: can you comment on why treatment with 5% N-9 results in lower pathological scores and neutrophil recruitment than 2% N-9?

This table gives the mean \pm SEM for pathological score and neutrophil infiltration for the three different concentrations of N-9 used:

	2% N-9	5% N-9	10% N-9
Pathology score Cervix	4.33 \pm 0.36	3.89 \pm 0.29	4.42 \pm 0.48
Neutrophil infiltr Cervix	2.75 \pm 0.75	1.33 \pm 0.33	3.75 \pm 0.75
Pathology score Vagina	4.08 \pm 0.37	3.56 \pm 0.95	4.42 \pm 0.34
Neutrophil infiltr Vagina	4.75 \pm 0.63	3.67 \pm 1.45	5.5 \pm 0.5

We can see what the reviewer perceives as a decreased score for 5% N-9, but by examining the standard errors of the mean associated with the means, it is apparent that these are overlapping ranges for the majority of comparisons and statistical interrogation does not support a hypothesis that the values for 5% are lower. Cervical neutrophil infiltration at 5% N-9 does appear lower and the observation

might be due to random type 2 variation, and not statistically different when multiple comparisons taken into account. It would be also difficult to formulate an explanation that would account for 5% N-9 not being as pathological as N-9 concentrations on either side of this (e.g. 2% and 10%). Of note, it has also been reported that in non-pregnant mice even a dose of 1% N-9 has been shown to disrupt the epithelia (doi: 10.1128/AAC.48.5.1837-1847.2004). Consequently, we believe that N-9 is a highly potent damaging agent across this whole range of concentrations. We elected to use the 10% dose in subsequent experiments as this resulted in a consistently significant increase in all scores compared to the vehicle treatment for both the cervix and the vagina. It is also pharmacologically relevant as it is within the concentration range that has been used in spermicidal agents (2-12%).

5. Figure 1: Can you please comment on why PBS could have an effect on pathological scores?

We appreciate the reviewer's comment regarding a potential effect by PBS. We did not include a 'no treatment control' in our experimental plan (in line with the 3 Rs principles – to do so would have been expensive in terms of animal numbers, yet yield little additional clinically relevant scientific information for our study). It is thus impossible from our data to say whether PBS inoculation (our vehicle control) has an effect on the normal epithelia or not.

Our scoring system has a baseline of 1 (i.e. no score can be lower than 1). Some variation in scores in control tissues would be expected and the scores for PBS treatments were 2.083 ± 0.37 for the cervix and 1.54 ± 0.34 for the vagina. They are thus not indicative of substantial damage. We have no evidence that PBS itself is toxic to epithelia. It is however feasible that some minor damage may result from i) irritation due to contact with the pipette tip during treatment or control administration ii) mechanical irritation or abrasion on inoculation with treatment or control iii) damage during tissue fixation, paraffin embedding, cutting and staining. Extreme care was taken to standardise procedures and ensure consistency between control and N-9 treatments, so there is no reason to expect process differences between control and N-9 treated tissues. We can therefore be confident that the PBS control is appropriate to evaluate N-9 effects on the epithelia that were due to the effect of the chemical itself, rather than due to the process of inoculation or diluent.

6. Figure 2: Can you please provide representative neutrophil IHC images for each of the groups.

We welcome the reviewer's request and attach images below. Should the reviewer and/or the editor wish, we are happy to include the representative images from each group either in the main manuscript or as a supplemental figure.

7. Supplementary Figure 4: there is a non-significant trend to increased time to delivery at higher N-9 concentrations (yet with reduced pup survival), can you explain this?

The values mean \pm SEM for delivery time are as below in the table:

	PBS	2% N-9	5% N-9	10% N-9	40% N-9
Time to delivery (hrs)	57.93 \pm 2	60.05 \pm 4.97	58.1 \pm 3.58	64.34 \pm 4.18	68.3 \pm 3.66

We appreciate the reviewer's impression that there is a trend towards delayed delivery for the highest concentrations of N-9 treatment. We have re-examined these values and confirm that the trend does not meet pre-specified statistical significance ($p < 0.05$) when analysed appropriately taking into account multiple

comparisons. Closer examination of the groups show delivery times for those treated with N-9 concentration of 2% and 5% groups are almost identical to the PBS group, and the 10% N-9 group is largely influenced by 1 mouse delivering at about 90 hours.

There may be a trend for the highest concentration group (40% N-9) to have delayed birth compared to PBS. As our hypothesis was that epithelial damage would induce PTB, our sample size was chosen to detect a difference of 12 hours i.e. we can exclude a difference of 12 hours, but are underpowered to exclude smaller differences. Nevertheless, delivery times of 57.93 hours (PBS) and 68.3 hours (40% N9) are both well within the normal limits of gestation for C57Bl/6 mice used (60 ± 3 hours based on extensive experience with similar models in our lab), and so the biological relevance of this difference in time to birth is questionable, even if it were statistically significant. Please note that the above variation values represent SEM and therefore no overlap does not guarantee statistical significance. Furthermore, 40% N9 was not used in subsequent experiments. We purposively included it as an extreme dose at the upper end of our dose response experiments, and it is a concentration much higher than would be used clinically. Multiple factors initiate parturition, including fetal cortisol, maternal drop in progesterone, oxytocin, prostaglandin F₂ α and other endocrine factors [doi's: 10.1677/joe.0.0420323, 10.1210/endo-107-1-155, 10.1126/science.277.5326.681, 10.1262/jrd.17033]. We believe that it would be beyond the focus of this manuscript to elucidate or speculate on a mechanism for the possible delay in delivery for animals treated with 40% N-9; and highlighting this possible non-significant trend within the manuscript would detract from our central hypothesis.

8. Was the Nano-luc expressing bacteria generated in this study? – if so, please can you give further information in the methods regarding the transposon mutagenesis.

The methods for transposon mutagenesis using pMT85 by us has been previously detailed in the Aboklaish et al., 2014 publication cited in the methods. However, we apologise for the lack of details regarding addition of NanoLuc genes to this vector. The methods have been amended to detail the vector modification and the full sequence of the resultant vector used to generate bioluminescent Ureaplasma has been deposited in GenBank (accession number MN275117) and is now appropriately referenced in the manuscript.

9. Line 175; one of the benefits of bioluminescent imaging is the ability to longitudinally image mice – do you have any data on the time course of bacterial infection?

We agree with the reviewer that bioluminescent imaging has some excellent advantages. However, we restricted imaging to a single timepoint for two main practical reasons. I) each imaging episode requires depilation and anaesthesia of the pregnant mice (with unknown extraneous effects) and ii) the time course model we

adapted from our previous LPS studies was only 48 hours from infection to sample collection, so there were minimal opportunities for imaging without undue stress on the pregnant animals (which itself may influence time to delivery). Further, as this manuscript represents the first of its kind for ascending *Ureaplasma* infection, we also embraced the expectation that several types of conformational methodology would have to be employed to validate the use of luciferase imaging (i.e. qPCR quantification of *Ureaplasma* from multiple sampling sites and bacterial titration of vaginal washes, amniotic fluid, and tissue samples) as well as collecting histological samples for grading. Furthermore, timed mating of initial batches of 30 mice generally resulted in 4 consecutive days of Caesarean delivery of proximal and distal pups with individual processing and sample collection for groups of 4-6 pregnant dams per day. Addition of an imaging step on day of delivery would delay timely completion of processing for each daily batch of mice and risked adding cumulative variation to length of time between infection and delivery and was therefore not performed.

10. Did you image your bacterial-treated mice at the time of administration of bacteria? This is necessary to prove that it's the cervical damage that leads to increased bacterial ascent.

In our preliminary studies we found that we could image the inoculum by adding substrate directly to the thawed NanoLuc-*Ureaplasma* aliquot tubes; however, subsequent attempts to get a sufficient image in vivo during the administration of the luminescent bacteria both vaginally, or under ultrasound-guided amniocentesis to a specific pup, were unsuccessful (See representative figures below). Therefore, during these studies we concentrated our efforts on trying to ensure minimum inoculum leakage occurred from the vagina following administration (which could not be achieved with concurrent/immediately subsequent in vivo imaging) to minimise received dose variation between infected mice. Further evidence of low visibility of sub-cervical NanoLuc luminescence is apparent in the images of infection at 24 hours, where the vaginal washes give evidence of consistent titre between all animals experimentally infected with *Ureaplasma* (irrespective of cervical challenge with N-9), but only gives a good signal for those animals with successful ascending infection. Whether this is due to overall differences in total *Ureaplasma* between ascending and sub-cervically confined infection or whether it is due to greater obstruction to the imaging camera between the vagina and the transdermal visualisation of the uterine compartment (or both), is unknown. However, as the amount of *Ureaplasma* required to image transdermally must therefore be much higher, we feel this adds additional evidence that the inoculum attains log-phase growth rapidly following inoculation (see reviewer 1 question 1).

(White band on each tube represents the adhesive label – liquid volume in all tubes is that the bottom). NanoLuc labelling in this image represents addition of the NanoLuc substrate as all bacteria contained the NanoLuc gene.

11. Line 187, do you have any representative images showing the uterine bulges/gestating pups? This is not clear in the provided images. Do you have any imaging from individual organs?

Given variation in position of the uterine horns when in situ, assigning pup order from an in vivo image (either bioluminescence or conventional greyscale ultrasound) is problematic, as pups/sacs may over lie one another. This is in contrast with the

nicely aligned bulges that can be seen in ex vivo experiments or when the mice are cut open under anaesthesia and have their uterine horns exteriorised for imaging. Another factor that needs to be taken into account is the position of the fetus and the placenta within the sac: a fetus with its back lying anteriorly might absorb a greater amount of bioluminescence.

In line with the 3R's we elected not to perform a laparotomy to our mice as we could obtain sufficiently clear images after the substrate injection. However, we attach some further images for the reviewer with our best interpretation of individual sacs. No individual organs can be distinguished from these images. As sterile separate processing of amniotic fluid and dissection of a range of placenta, uterine tissue and organs occurred for each individual pup delivered (noting their order/distance from the cervix) with time restrictions for flash-freezing samples for subsequent extraction for RNA analysis: there was insufficient time to have included attempted individual imaging of pups or dissected organs without compromising our other planned analyses.

12. Figure 5; there are significant amounts of Ureaplasma in amniotic fluid, yet non-significant differences in Ureaplasma in fetal lungs, can you please comment on this? Do you have any evidence of bacterial infection in live or dead pups?

Our expectations mirror those of the reviewer: We had expected a significant infection of the fetal lungs by Ureaplasma. The literature leads us to expect a significant amount of exchange (or “breathing”) between the intrapulmonary space and the amniotic cavity, although it is unclear if this is consistent in all strains of mice. Our findings may reflect the fact that only 48 hours had elapsed between vaginal inoculation and delivery of the pups, and Ureaplasma infection was only on the cusp of establishing infection in the lungs (the volume of amniotic fluid relative to the weight of lung tissue measured out for extraction would be quite small). However, we cannot exclude other mechanisms at play, and a failure for Ureaplasma to establish experimental lung infection during intra-uterine administration of C57Bl/6 fetal mice has been previously reported (von Charmier et al doi: 10.1371/journal.pone.0044047).

Do you have any evidence of bacterial infection in live or dead pups?

Our animal licence supported a mechanistic study of bacterial infection and the effect on gestation, and we did not assess degree of fetal/neonatal injury or pup survival in response to UP infection.

Except for the 13% and 28% of dams that spontaneously delivered preterm in the UP and UP+N-9 groups respectively (see below), we surgically delivered near-full term gestation fetuses under terminal maternal anaesthesia with fetuses immediately euthanized, as per requirement of our animal project licence. We were therefore unable to assess pup survival at delivery. We have I) adjusted the methods section to make this clearer and II) have replaced “pup” with “fetus” throughout the manuscript, to help clarity (In line with reviewer 2 comment xx). However, we noted no signs of resorption or friability of organs during sample collection, so there was no clear signal that *Ureaplasma* infection impacted in utero survival.

Our project licence also required us to cull any surviving preterm born pups on discovery. 63/98 pups were discovered alive during twice daily cage checks. CCTV evidence suggested a proportion of those pups found dead or partially eaten were initially mobile and could therefore be presumed to be live born as well. This is again suggestive that *Ureaplasma* was not substantially affecting in utero survival. Further indirect support for this is found in Table 1. This shows a close correlation between mRNA expression of *Ureaplasma* genes (*ureC*) and cytokine mRNA levels, suggest coincident viability of both the bacteria and the fetus respectively, even under high bacterial load.

We agree that it would be interesting in future studies to study the effect of *Ureaplasma* on ex utero survival and pup behaviour, but this was outwith the scope of the current study.

13. How does the N-9 damage the cervical barrier? Do you have any evidence of pro-inflammatory change or epithelial barrier integrity/permeability?

N-9 is a non-ionic surfactant. As such, its natural affinity to the lipids of the cell membrane allows it to integrate in the cell membrane, hold the lipids in suspension and eventually detach them and lyse the membrane, leading to cell death. Its effects on epithelial surfaces have been reviewed before (doi: 10.1097/01.qai.0000159671.25950.74). Briefly, N-9 has been shown to be cytotoxic against epithelial cells in vitro. In lower subtoxic doses, it can trigger the release of proinflammatory cytokines such as IL1- β and IL-8. Similar findings were also observed ex vivo and in vivo, mostly in rabbits but also in rats, mice and monkeys.

We have preliminary in vitro data to assess N-9 damage to a human immortalised cervical epithelial cell line (End1/E6E7). We found a time and dose dependent cytotoxic effect of N-9 on these cells and we also found a severe disruption of the permeability of an endocervical cell monolayer. Please see figures below. Please also note the in vitro doses are in the magnitude of 100- to 1000-fold lower compared to the doses routinely used in vivo.

Treatment of endocervical cells (End1/E6E7) with N-9 results in decreased cell metabolic activity in a dose- and time-dependent manner. End1/E6E7 cells were incubated with various N-9 concentrations (0, 2, 4, 8, 16, 32, 64, 128, 256 and 512 $\mu\text{g/ml}$) for 30min, 1h, 2h, 4h and 24h. Metabolic activity indicating cell viability was determined by an MTT assay and is expressed relative

to that of the untreated cells. Four independent experiments were conducted. Error bars indicate *SD*.

Medium and high doses of N-9 result in compromised epithelial barrier function of endocervical (End1/E6E7) cells. Confluent End1/E6E7 cells were seeded at 500,000 cells per insert for 72 hours in growth medium. Following this period, the monolayers were treated with different concentrations of N-9 diluted in growth medium (0 µg/ml, 3 µg/ml, 10 µg/ml, 30 µg/ml and 100 µg/ml) for 2 hours. A FITC-Dextran solution was added to the inserts for 1 hour and the fluorescence intensity (F. I.) of the receiver wells was measured using CLARIOstar multimode microplate reader. High N-9 concentrations (30 and 100 µg/ml) resulted in higher F. I. of the respective receiver wells compared to low N-9 concentrations (0, 3 and 10 µg/ml), indicating increased epithelial permeability. Five independent experiments were conducted. Error bars indicate *SD*.

Neutrophil recruitment was increased in the N-9 group, yet following bacterial administration, more bacteria survive – can you comment on this?

High concentrations of *Ureaplasma* coincident with large numbers of infiltrating pulmonary neutrophils is a common clinical finding. The authors of a recent publication by Gomez-Lopez et al., (2017 *Am J Reprod Immunol.* 78(4). doi: 10.1111/aji.12723) came to the conclusion that in contrast to intra-amniotic infection by *Streptococcus agalactiae*, *Bacteriodes fragilis*, *Prevotella* spp., and *Gardnerella vaginalis* that “amniotic fluid neutrophils seem to display a delayed ability to phagocytize *Ureaplasma urealyticum* and *Escherichia coli*”. Combined with the well-

characterised TLR2/6 and 9 stimulation by *Ureaplasma parvum* (See discussion lines 383-388), which includes induction of potent neutrophil chemotaxins IL-8 in humans (the murine equivalent being CXCL-2 as found elevated in our manuscript-Figure 6), we would expect higher infiltration of neutrophils at sites of active *Ureaplasma* infection with poor capacity to stem the bacterial infection.

Reviewer #2 (Remarks to the Author):

In this manuscript the authors developed an ascending infection model in mice using a human opportunistic pathogen, *Ureaplasma parvum* combined with cervical/vaginal injury induced with Nonoxynol-9. The authors provide evidence that cervical injury facilitates ascending infection with *U. parvum* and that this is accompanied by preterm labor. A strength of the paper is that the authors used a controversial pathogen that is translationally relevant to human disease. The controversy stems from the fact that infection does not always induce adverse pregnancy outcome. Herein, the authors show that cervical injury is critical for *U. parvum* ascending infection and inflammation linked to preterm birth. While this point is of interest to the field of maternal-fetal medicine, there are some concerns that need to be addressed.

Point 1) Introduction: The key feature in this paper is the interaction of Nonoxynol-9 and *U. parvum* on pregnancy outcome.

The authors did not provide a careful explanation of *U. parvum* as an opportunist and the confounding issues with its role in adverse pregnancy outcome. Doing so would actually raise the impact of their paper. I recommend the authors deemphasize *U. parvum* in the first paragraph and instead summarize the current views regarding the mechanics of preterm birth – including the reemerging view that decidual/myometrial inflammation is the main trigger for preterm labor (this is relevant to some of data presented).

Following the paragraph after the cervix I would add a third paragraph that supports the rationale for using *U. parvum* in this study, including that it is viewed as an opportunistic pathogen and that not all upper reproductive tract infections end up with complications (this includes human studies). Moreover, if the authors do a careful review of *U. parvum* related disease, they will find that tissue injury is important in how this organism enhances disease. This actually strengthens the rationale for using this organism in the first place. As an example, the authors cited Novyi et al, who demonstrated that intraamniotic inoculation of *U. parvum* in pregnant macaques induced preterm labor. While this is true, this study was later contradicted by a paper published by Kallapur's research group that showed that the same strain of *U. parvum* administered in the same way in macaques only induced a

nominal amount of inflammation with no induction of preterm labor. What was the difference

between these two studies? In Novyi's study, macaques had prior surgery with implantation of electrodes to monitor labor, whereas Kallapur's research group did not. Perhaps the surgery induced enough of a stimulus that enhanced the host immune response to *U. parvum*, and this led to preterm labor. There are other examples in the literature to support this view.

We have re-arranged the introduction as requested by the reviewer to highlight the role of *Ureaplasma* as an opportunist. We were unable to find any publication by Kallapur et al. per se in PubMed. However, we did find a reference including this author showing no preterm birth in a manuscript by Boonkasidecha et al., referring to groups of pregnant macaques to LPS, IL-1 antagonists and *Ureaplasma*, that mentions no preterm labor in any of these groups and we have used this as the citation in the manuscript. Nevertheless, the reviewer's thoughts on the differences between these studies are interesting and supports a hypothesis that more than one 'insult' potentiates the likelihood of preterm birth. Our findings are in agreement with this idea. However, we recognise that all animal models have different strengths and weaknesses, and we have chosen to restrict discussion to the most relevant literature to our model, and corroborating clinically relevant findings.

Point 2) Methods: the current view is to provide adequate reporting of live animal studies. Please refer to ARRIVE Guidelines for details, most scientific journals provide a link to this. Key details that need to be added in the methods include animal housing. Were there specific pathogen free animals? What were the housing conditions? Were they maintained under barrier maintenance or conventional housing? Necropsy procedures are not clearly described and are confusing. It sounds as if the pregnancy was terminated before actual labor, if so what was the final gestation day? Technically, these are not pups but fetuses. What portion of the myometrium was isolated, the entire uterus or only the placental bed? The placental bed (myometrial portion in direct association with the placenta) has a unique composition of leukocytes not found in the uterus that is not associated with fetal tissue. This needs to be clarified

We thank the reviewer for this. We have updated the methods section in accordance with the suggested guidelines (relevant fields from the manuscript below). We also provide a schematic figure of the experimental design for the reviewer to see below. Finally, we now uniformly use the term fetus throughout the manuscript as we do agree it is more accurate.

Animal studies

All animal studies were performed under UK Home Office License 70/8927 (PPL) to J. E. N. and in accordance with the UK Animals (Scientific Procedures) Act of 1986.

Pathogen-free virgin female C57Bl/6J mice were purchased from Charles River Laboratories (Margate, UK) at 6-8 weeks of age. They were housed under tightly regulated temperature (19-23°C), humidity (55%) and light (12 h dark/light cycles) conditions. Mice were acclimatised for 10 days and then time mated. Confirmation of successful pregnancy was equated to vaginal plug presence at day1 of pregnancy. Five mice were housed in one conventional cage with regular bedding changes and 24-hour access to food and water until the day they would undergo an experimental procedure. Following the procedure, mice were housed in individual conventional cages in an isolated room under CCTV monitoring and under the same husbandry conditions. Pathogen-free status of the housing facility is confirmed by regular cleaning/sterilisation of all rooms with quarterly harvest of sentinel animals in each housing area to confirm absence of specified pathogen transmission. All proposed experiments and procedures were reviewed and approved by the Named Training and Competency Officer and Named Veterinary Surgeon of the University of Edinburgh. No unexpected adverse events were noted ensuring no breach of the Project Licence Protocols.

Mouse model of cervical damage and ascending infection

In the afternoon of day 16 gestation, mice were anaesthetized and received either 10% v/v N-9 or PBS as above, monitored for leakage during recovery and returned to individual cages. In the morning of day 17 gestation mice were re-anaesthetised and received a further vaginal inoculation of either UP-Luciferase (40µl aliquot of 10⁸ CCU/ml in USM) or 40µl USM alone. Mice were randomly assigned to each group. To reduce the likelihood of cross contamination, the group of mice that was treated with UP-Luciferase was handled last. Following successful recovery from anaesthesia, mice were put in individual cages to monitor the time to delivery by continual CCTV (including recording for accurate determination of emergence of first pup). Preterm delivery was determined as birth occurring <48 h post-infection (or mock-infection) as normal term delivery is known to be approximately 60 h from this point. Mice that have not delivered by the 48-h mark after administration of UP-Luciferase (completed D18 of gestation) were deemed as term and were sacrificed for collection of antepartum tissues. During this period, they also received twice daily physical health checks for signs of potential adverse effects.

Point 3) Methods: how were lesion scores actually obtained? A software program was mentioned but the actual procedure used was not described. How many sections from each animal were examined? Were multiple sections evaluated? How did the software calculate percentages that were later assigned a score? Was the analysis done blinded to treatment?

We appreciate the reviewer's request and have therefore updated the relevant sections in our Methods with all the information requested by the reviewer. We also attach a schematic figure of our tissue processing and analysis strategy for the reviewer. Please also note that the weighted Cohen's kappa for each figure that are relevant for have been added to the main text in the Results sections.

Tissue collections

To accurately assess tissue damage, we deployed the following strategy for processing and analysing our samples: 3 consecutive 5µm-thick longitudinal sections (used for histology, Ly6G and Ki67 IHC respectively) from 3 different levels of the lower reproductive tract were processed, each level 50µm deeper than the preceding one. Subsequently obtained scores for each mouse represent the average of the scores for the 3 different levels.

Histological imaging analysis

Imaging of the samples was performed on a Zeiss Axio Scan.Z1 Slide Scanner (Carl Zeiss AG, Oberkochen, Germany). Imaging analysis was performed using Zeiss ZEN Blue software (Carl Zeiss AG). This allows for individual operator-led analysis of the images obtained by the Slide Scanner. Specifically, two independent assessors that were blinded to treatment allocation evaluated all images based on the scoring systems described in the respective Results sections. Individual scores for each mouse represent the average of the scores assigned by the assessors. Inter-rater reliability was assessed using Weighted Cohen's kappa. Please note that all the calculated Cohen's kappas were above 0.75 indicating a substantial assessor agreement.

Point 4) Results: Animal numbers should be clearly stated in each experiment. It is hard to tell in some of the figures what the actual sample sizes are for some of these experiments. Please provide animal numbers in figure legends

N numbers now included in all figures.

Point 5) If infected animals were allowed to go to term, what was the survival rate among pups?

We thank the reviewer for highlighting this area of confusion. As correctly identified, all animals that did not deliver preterm (13% of UP only and 28% of N-9+UP), were processed near-term (end of day 18 gestation) by Caesarean delivery of late gestational age fetuses under terminal anaesthesia. We have clarified this in the methods section. It is also clarified in detail in our response to a comment by reviewer 1 so to avoid repetition, we direct the reviewer to our answer to the 2nd part of point 12 above. Consequently, we were unable to assess fetal viability at time of delivery. However, there were no signs of resorption or friability of organs during sample collection, anecdotally suggesting all foetuses were alive prior to delivery procedure. Higher cytokine mRNA levels in fetuses with the highest ureC Ureaplasma mRNA (Table 1) also suggest full viability of foetuses with high bacterial loads in utero.

Of the 13% of UP treated and 28% of UP+N-9 treated dams that had spontaneous preterm birth, a majority of these were discovered live born and our animal license required their immediate culling. As our study was designed to assess mechanistic ascending bacterial infection and not fetal injury, we were unable to perform any studies regarding in utero Ureaplasma infection on neonatal morbidity and mortality.

Point 6) The text in Figure 7 is unreadable. Font sizes need to be increased so that the figure is legible.

We have increased font sizes in the figure in line with the reviewer's request.

Point 7) Discussion: line 315, The statement “Our findings confirm the role of the vaginal commensal UP in causing pregnancy complications by demonstrating its potential to ascend to the gravid uterus and induce inflammation and preterm birth” I believe this statement needs to be modified to state the importance of N-9 in contributing to this since myometrial inflammation and a significant increase in preterm birth was not observed in the UP only group. Although N-9 by itself did not induce preterm labor or inflammatory response in the upper reproductive tract, the initiation of inflammation cascades by the tissue damage may be a contributing factor (think of DAMPS). Somewhere in the discussion this should be addressed especially since many studies have used bacterial LPS as a substitute for infection and get more dramatic results than *U. parvum* infection alone.

We agree with the reviewer and have made the following alterations to the discussion to address these points:

Discussion Paragraph 1

We have shown that cervical epithelial damage facilitates ascending UP infection into the uteri of pregnant mice, with accompanying preterm birth and elevation of pro-inflammatory cytokines in the myometrium, fetal membranes and placenta. Our findings further support the role of the vaginal commensal UP in causing pregnancy complications by demonstrating its potential to ascend to the gravid uterus and induce intrauterine inflammation. Importantly, when preceded by cervical damage, UP infection elicits a more robust intrauterine inflammatory response which significantly increases the risk for PTB. This synergistic effect between the two insults is in line with current consensus towards PTB being a multifactorial syndrome (Romero et al., 2006). Our results also highlight the importance of the cervical epithelium in providing a protective barrier against infections in pregnancy, and provide additional rationale for the mechanisms underlying previous associations between excisional treatment for Carcinoma in Situ (CIN) and PTB. Cone biopsy or Large Loop excision of the Transformation Zone (LLETZ) involve removal of part of the cervical epithelium and underlying stroma and are associated with high risk of PTB (Kyrgiou et al., 2006) (Bruinsma and Quinn, 2011). Increased risk of preterm birth with larger or repeated excisions has previously been reported (Jakobsson et al., 2009), but most studies have focused on the mechanical importance of the cervix for providing structural support, ignoring the potential role of the epithelium (Stock and Norman, 2012). Our studies provide empirical evidence of the importance of epithelial integrity.

Discussion Paragraph 3

Our data confirm that N-9 damages the cervical epithelium during pregnancy, but when investigated alone it did not induce preterm birth or pup survival. This suggests that cervical epithelial damage in isolation is insufficient to cause PTB. However, ascending vaginal tract infection and subsequent inflammation are the most common cause of PTB. We found a significant increase in both *in vivo* bioluminescent imaging of ascending UP infection and increased titres of UP (both in magnitude and frequency of infection reaching mouse fetuses more distal from the cervix) following N-9 cervical epithelial damage. This demonstrates a direct relationship between cervical integrity and ascending infection. We also found a direct correlation in both PTB in mice with ascending infection and significant relationship between UP presence and increased TNF α IL1- β CXCL-1 and CXCL-2 cytokine expression in fetal membranes, placenta and the myometrium. This is a novel finding that is of direct translational importance to the human pregnancy, as *Ureaplasma* spp are the bacteria most commonly implicated in human PTB (DiGiulio, 2012). Our results closely mirror the findings of two recent papers both suggesting that cervical insults, be it either defective epithelial differentiation (Akgul et al., 2014) or a viral infection (Racicot et al., 2013), significantly increase the risk of PTB followed by vaginal *E. coli* administration. Whether the observed increase in PTB rates is solely due to the highest rates of infection and higher bacterial titres when there is prior cervical damage or whether cervical damage itself actively contributes to this phenotype, remains to be seen. It is biologically plausible that the influx of neutrophils into the cervical stroma caused by N-9 at least partly contributed to a pathological premature cervical remodelling as previously described (Holt et al., 2011). Even if not sufficient to induce PTB, it could have accelerated the process driven by UP-induced intrauterine inflammation. This further highlights intrauterine inflammation as an essential component of infection-mediated PTB. Indeed, intrauterine administration of LPS first described by Elovitz et al. consistently induces PTB and is now routinely used in PTB mouse models (Elovitz et al., 2003). However, our group has previously published that intravaginal administration of LPS alone could not induce PTB (Rinaldi et al., 2015).

REVIEWERS' COMMENTS:

Reviewer #2 (Remarks to the Author):

This is a revised manuscript from a previous submission where the authors describe a novel murine model for the studying the interaction of cervical/vaginal injury and ascending infection with the human urogenital opportunist, *U parvum*.

The authors have adequately addressed my concerns with the previous submission.